# A Unified Framework for Fair Graph Generation: Theoretical Guarantees and Empirical Advances

**Zichong Wang[1]**    **Zhipeng Yin[1]**    **Wenbin Zhang[1]** *
[1] Florida International University, Miami, FL, United States
`{ziwang, zyin007, wenbin.zhang}@fiu.edu`

## Abstract

Graph generation models play pivotal roles in many real-world applications, from data augmentation to privacy-preserving. Despite their deployment successes, existing approaches often exhibit fairness issues, limiting their adoption in high-risk decision-making applications. Most existing fair graph generation works are based on autoregressive models that suffer from ordering sensitivity, while primarily addressing structural bias and overlooking the critical issue of feature bias. To this end, we propose FairGEM, a novel one-shot graph generation framework designed to mitigate both graph structural bias and node feature bias simultaneously. Furthermore, our theoretical analysis establishes that FairGEM delivers substantially stronger fairness guarantees than existing models while preserving generation quality. Extensive experiments across multiple real-world datasets demonstrate that FairGEM achieves superior performance in both generation quality and fairness.

## 1 Introduction

Graph learning has become increasingly important due to the ubiquity of graph-structured data across various domains, such as social networks [1], recommendation systems [2], and financial markets [3]. Among the important tasks in graph learning, graph generation stands out for its ability to create synthetic graph data that serves crucial purposes such as data augmentation [4], anomaly detection [5], and enabling privacy-preserving data sharing [6]. Essentially, graph generation models aim to capture the underlying data distribution and produce novel graph samples that maintain the statistical properties of the original data [7]. For instance, banks may employ graph generation models to construct synthetic applicant interaction networks based on historical loan approval data, allowing them to share insights with third-party testing agencies without exposing sensitive information [8].

Despite their significant success, existing graph generation models often neglect the crucial issue of fairness, which limits their adoption in high-risk decision-making scenarios, such as healthcare [9], credit scoring [10], and crime prediction [11], where biased graph generation can perpetuate or amplify social inequities. Returning to the banking example, the generated graphs may disproportionately create more connections among individuals from the same sensitive subgroup, further segregating the representations of nodes belonging to different sensitive subgroups. This leads to an over-association of downstream tasks with sensitive attributes, thereby reinforcing biases and amplifying discrimination in high-risk decisions (*e.g.*, loan decision), raising serious ethical issues.

There has thus been growing interest in exploring fairness in graph generation tasks, with preliminary efforts primarily focused on mitigating structural bias in graphs [6, 12, 13, 14, 15]. These works aim to address disparities in connection patterns across sensitive subgroups. However, they largely overlook *feature bias*, disparities that emerge when the generated node features differ systematically across subgroups. For example, the generated graph might show male nodes with higher incomes

---

*Corresponding author.

than female nodes, leading to biased downstream applications where models trained on this synthetic data learn to expect higher incomes from male applicants. Furthermore, existing fair graph generation models typically belong to autoregressive models that construct graphs incrementally [14]. These autoregressive models exhibit inherent limitations, including ordering sensitivity, where the generated graphs heavily depend on arbitrary node orderings, and difficulty capturing comprehensive global structural patterns efficiently [16].

To address these drawbacks, this paper explores fair one-shot graph generation models that treat graph components holistically, generating entire graphs simultaneously with inherent node permutation invariance, enabling better representation of global graph structures [17], while tackling both structural and feature biases. To achieve this, several challenges need to be addressed: **i) Difficulty in preserving natural attribute differences while removing unfair disparities.** When generating node features, certain attributes should naturally vary across sensitive groups, while others should not show group-based disparities. The fundamental challenge lies in developing methods that can reliably distinguish between these two types of features from the data. Without robust techniques for this identification, one-shot generation models will either inappropriately flatten natural variations or perpetuate unfair disparities, compromising either data realism or fairness. **ii) Difficulty in coordinating fairness across interconnected graph components.** One-shot models generate node features and structural patterns simultaneously, creating complex dependencies. Without coordinated fairness mechanisms, addressing bias in one component can inadvertently exacerbate bias in another. This interdependence can lead to models that appear fair in isolated metrics but still produce systematically biased results when evaluated holistically. **(iii) Difficulty in formulating and proving theoretical guarantees.** In one-shot graph generation, the entire structure and node features emerge from random noise in a single pass, which requires any fairness constraint to be incorporated from the outset and remain valid through the entire noise-to-graph transformation. As a result, proving formal fairness bounds under such a one-shot framework requires carefully designed regularizers and rigorous analysis that can guarantee fairness in the final generated graph.

To tackle the above challenges, this paper introduces a novel framework, ***Fair Graph gEnerative Modeling*** (FairGEM), which achieves fair graph generation through specialized regularization of both structural and feature components in spectral diffusion models. *To the best of our knowledge, this is the first work that theoretically grounds fair graph generation to simultaneously address both structural and feature biases while avoiding node ordering dependencies.* Specifically, we mitigate graph structural bias by developing a fairness regularizer that quantifies and minimizes discrepancies between intra-group and inter-group edge reconstructions, while addressing node feature bias through a disentanglement-guided strategy and a fairness regularizer that selectively enforces fairness on sensitive-irrelevant features. Furthermore, we establish theoretical guarantees for our framework, providing upper bounds on bias propagation and demonstrating how our method directly improves fairness in downstream tasks. Our main contributions can be summarized as follows:

- **Theoretical analysis.** We establish the first theoretical foundation for fairness-aware graph generation that works without sequential node ordering constraints. Our analysis provides upper bounds on both structural and feature bias propagation, demonstrating how these biases impact downstream task disparities.

- **Novel Framework.** We establish a general framework for fair graph generation through FairGEM, which introduces two specialized regularizers: one that minimizes discrepancies between intra-group and inter-group edge reconstructions, and another that disentangles and selectively regularizes sensitive-unrelated features.

- **Extensive Experiment Evaluation.** We conduct extensive experiments to evaluate by comparing it with the state-of-the-art methods across four real-world datasets, achieving a significant improvement in fairness metrics while maintaining generation quality.

## 2 Related Works

**Graph Generation Models.** Generating synthetic graphs has been extensively explored through deep generative models, categorized mainly into autoregressive and one-shot methods [17]. Specifically, autoregressive methods, including those based on recurrent neural networks [18, 19], and reinforcement learning [20, 21], construct graphs incrementally by adding nodes and edges in sequence. While these approaches can capture complex structural patterns, they suffer from inherent limitations such

as sensitivity to node ordering and difficulty in modeling global graph properties efficiently. To address these limitations, one-shot graph generative models construct edges simultaneously, typically employing Variational Autoencoders [22, 23, 24], Generative Adversarial Networks [25, 26, 27], and spectral diffusion models [28, 29], offering advantages such as node permutation invariance and holistic representation of graph structures. Despite these advances in synthetic graph generation quality across both paradigms, fairness concerns remain largely unexplored in the literature, which severely limits their applicability in high-risk decision-making scenarios, thus creating an urgent need to develop fairer graph generation methods.

**Fairness-aware Graph Generation.** There is a growing effort in the research community to develop fair graph learning models addressing biases in applications [30, 31, 32, 33, 34, 35, 36, 37], however, most existing fairness-aware methods primarily focus on classification tasks, leaving the fairness challenges in graph generation largely unexplored [38]. Recently, a small number of works [12, 39, 40] have begun to investigate fairness specifically within graph generation, primarily adopting autoregressive methods for tasks such as fair link prediction and fair structural generation. Fair link prediction methods aim at unbiased inference of edges between nodes; for instance, FAIRLP [13] adjusts the training graph to balance intra-group and inter-group link distributions, enhancing representation fairness. Fair graph structural generation methods, on the other hand, address fairness at the structural level by reducing distributional disparities between generated graphs and the original graph across demographic subgroups. For instance, FairGen [40] introduces parity constraints to minimize subgroup reconstruction differences. However, these existing approaches focus on structural fairness, neglecting biases in node feature generation, highlighting the need for comprehensive fairness solutions in synthetic graph generation tasks. In addition, these autoregressive models inherently suffer from ordering sensitivity, meaning generated outcomes can vary significantly with different node orderings, inadvertently causing biases.

In contrast to existing work, this paper proposes a fair one-shot graph generation model that addresses both graph structural bias and feature bias, with its design informed by theoretical analysis. By leveraging the holistic nature of one-shot generation, our approach can simultaneously optimize fairness without the ordering sensitivity that plagues sequential methods. Additionally, our bias mitigation approach is flexible, allowing it to be applied in training both link prediction models and generative models to create fair synthetic graphs.

## 3 Notation

Given an attributed graph $\mathcal{G} = (\mathcal{V}, \mathcal{E}, \mathbf{X})$, where $\mathcal{V} = \{v_1, v_2, \ldots, v_n\}$ represents the set of nodes and $\mathcal{E} \subseteq \{\{v_i, v_j\} \mid v_i, v_j \in \mathcal{V}\}$ denotes the set of undirected edges. The node feature information associated with the graph is represented by a feature matrix $\mathbf{X} \in \mathbb{R}^{n \times d}$, where each node $v_i$ corresponds to a $d$-dimensional feature vector $\mathbf{x}_i \in \mathbb{R}^d$. Graph connectivity is captured by the adjacency matrix $\mathbf{A} \in \{0, 1\}^{n \times n}$, where $\mathbf{A}_{i,j} = 1$ indicates that nodes $v_i$ and $v_j$ are connected by an edge, and $\mathbf{A}_{i,j} = 0$ otherwise. We assume each node is associated with a binary sensitive attribute $s_i$, represented by the vector $\mathbf{S} \in \{0, 1\}^{n \times 1}$, where $s_i$ denotes the sensitive attribute for node $v_i$. The node set can thus be partitioned into two groups based on these sensitive attributes: the deprived group $S_d = \{v_i \in \mathcal{V} \mid s_i = 0\}$ (*e.g.*, female), and the favored group $S_f = \{v_i \in \mathcal{V} \mid s_i = 1\}$ (*e.g.*, male). Additionally, each node $v_i$ carries a binary ground-truth label $y_i \in \{0, 1\}$ representing the outcome of interest, such as approval ($y_i = 1$) or rejection ($y_i = 0$). Predicted outcomes from the model are indicated as $\hat{y}_i$.

## 4 Methodology

This section introduces FairGEM, a novel framework designed to achieve fair graph generation by mitigating biases that emerge during the diffusion process. Specifically, in Section 4.1, we review the standard score-based graph diffusion model and identify two key sources of bias: structural bias in graph structural generation and feature bias in node feature generation. Section 4.2 presents our method for addressing graph structural bias by introducing a fair graph structure generation regularizer. Section 4.3 describes our approach for tackling node feature bias using a disentanglement-guided method, which separates sensitive-related from sensitive-irrelevant features, enabling targeted fairness regularization. In addition, we provide theoretical insights and guarantees regarding the effectiveness of our approach in mitigating bias in graph generation.

## 4.1 Inspection Biases in Graph Generation Process

We begin by examining the root causes of bias in graph generation, establishing a clear foundation for developing fair graph generative models. Understanding these causes requires a closer look at how such models are typically constructed, most aim to learn the joint distribution of node features ($\mathbf{X}$) and graph structure ($\mathbf{A}$). To effectively capture this joint distribution, recent work has increasingly adopted score-based diffusion modeling, a powerful strategy that transforms complex data distributions into simpler ones through the controlled addition of noise [41]. Specifically, diffusion models leverage stochastic differential equations (SDEs) to systematically perturb the original data distribution over continuous timesteps until it approximates a simple prior distribution, then learn to reverse this process. Mathematically, this graph diffusion process can be expressed as:

$$d\mathbf{X}_t = \mathbf{f}^{\mathbf{X}}(\mathbf{X}_t, t)dt + \sigma_{\mathbf{X},t}d\mathbf{B}_t^{\mathbf{X}}, \quad d\mathbf{A}_t = \mathbf{f}^{\mathbf{A}}(\mathbf{A}_t, t)dt + \sigma_{\mathbf{A},t}d\mathbf{B}_t^{\mathbf{A}} \tag{1}$$

where the drift functions $\mathbf{f}^{\mathbf{X}}(\cdot, t)$ and $\mathbf{f}^{\mathbf{A}}(\cdot, t)$ control the deterministic transformations of node features and graph structures, respectively. $\mathbf{X}_t$ and $\mathbf{A}_t$ denote the random states of node features and the adjacency matrix at time $t$. The stochastic terms governed by $\sigma_{\mathbf{X},t}$ and $\sigma_{\mathbf{A},t}$ determine the intensity of random noise introduced via Brownian motions $(\mathbf{B}_t^{\mathbf{X}})$ and $(\mathbf{B}_t^{\mathbf{A}})$.

To reverse this diffusion process and generate realistic graph samples from noisy data, score-based models estimate the score function, which represents the gradient of the log probability density at various noise levels. This function is approximated using a neural network trained via the denoising score matching objective:

$$\begin{cases} \mathcal{L}_{\mathbf{X}}(\theta) \triangleq \mathbb{E}_{\mathbf{G}\sim\text{Unif}(\mathcal{Z})}\mathbb{E}_{\mathbf{X}_t|\mathbf{G}}\|z_\theta(\mathbf{X}_t, \mathbf{\Lambda}_t) - \nabla \log p_{t|0}(\mathbf{X}_t|\mathbf{X}_0)\|^2 \\ \mathcal{L}_{\mathbf{A}}(\phi) \triangleq \mathbb{E}_{\mathbf{G}\sim\text{Unif}(\mathcal{Z})}\mathbb{E}_{\mathbf{\Lambda}_t|\mathbf{G}}\|z_\phi(\mathbf{X}_t, \mathbf{\Lambda}_t) - \nabla \log p_{t|0}(\mathbf{\Lambda}_t|\mathbf{\Lambda}_0)\|^2 \end{cases} \tag{2}$$

where $\mathbf{G} \sim \text{Unif}(\mathcal{Z})$ represents uniform sampling from the training set; $t \sim \mathcal{U}(0, 1)$ indicates sampling a timestep from $[0, 1]$; $(\mathbf{X}_t|\mathbf{X}_0)$ and $(\mathbf{A}_t|\mathbf{A}_0)$ denote noisy versions at time $t$; $z_\theta$ and $z_\phi$ are neural networks predicting conditional scores; $\nabla \mathbf{X} \log p(\mathbf{X}_t|\mathbf{X}_0)$ and $\nabla \mathbf{A} \log p(\mathbf{A}_t|\mathbf{A}_0)$ are the true conditional scores; and $|| \cdot ||_F$ is the Frobenius norm.

However, this diffusion-based generation inherently propagates and amplifies existing biases present in the original data distributions [42]. Specifically, in the forward SDE for $\mathbf{A}_t$, biased patterns such as denser connections among nodes sharing sensitive attributes (*e.g.*, gender) remain largely intact because the noise addition step ($\sigma_{\mathbf{X},t}$, $\sigma_{\mathbf{A},t}$) does not alter the *relative* densities of these structures, which continue to dominate the evolving distribution. Similarly, in the SDE for $\mathbf{X}_t$, feature biases, manifested as distributional disparities across sensitive groups, persist through the forward diffusion stage, since added noise minimally shifts those underlying imbalances. Subsequently, during the reverse diffusion process, the score function $\nabla \log p(\mathbf{X}_t, \mathbf{A}_t)$ (learned via mean-squared error minimization) directs generated samples toward regions of higher data density. Because these regions correspond precisely to biased modes, the generative model inherently intensifies structural and feature biases. In summary, the optimization process of the score function, typically guided by mean-squared error loss, inherently prioritizes more frequent and dominant biased patterns in the training data. This weighting implicitly assigns greater importance and thus greater accuracy to these biased modes. In turn, this amplifies existing disparities, with minority groups and less frequent patterns disproportionately neglected.

## 4.2 Mitigation Graph Structural Bias in Graph Generation Process

Guided by the bias analysis, two fairness regularizers are proposed to explicitly address biases in graph structure and node features. The first, a structural fairness regularizer, is designed to mitigate structural bias. It does so by quantifying the discrepancy between reconstruction errors on intra-group versus inter-group edges, as formally defined in Definition 4.1.

**Definition 4.1 (Graph Structure Information Generation Bias)** *Given $\mathcal{G}$ with $\mathbf{A}$ and $\mathbf{S}$, the bias in graph structure information generation is defined by the disparity in how a generative model reconstructs connections between nodes from the same subgroup defined by sensitive attribute versus connections between nodes from different subgroups. Specific to spectral diffusion framework, where*

*a fixed initial eigenvector matrix $\mathbf{U}_0$ guides the evolution of the adjacency matrix as $\mathbf{A}_t = \mathbf{U}_0 \Lambda_t \mathbf{U}_0^\top$ through a Gaussian process, this bias at diffusion time $t$ can be formally quantified as:*

$$\Phi_{struct}(\Lambda_t) = (E_{intra}(\Lambda_t) - E_{inter}(\Lambda_t))^2 \tag{3}$$

*where $E_{intra}(\Lambda_t)$ and $E_{inter}(\Lambda_t)$ represent the reconstruction errors for intra-group and inter-group edges, respectively:*

$$E_{intra}(\Lambda_t) = |\mathbf{P}_{intra} \odot (\hat{\mathbf{A}}_t - \mathbf{A})|_F^2, \quad E_{inter}(\Lambda_t) = |\mathbf{P}_{inter} \odot (\hat{\mathbf{A}}_t - \mathbf{A})|_F^2 \tag{4}$$

*where $\hat{\mathbf{A}}_t$ represents the reconstructed adjacency matrix at time t, $\odot$ denotes the Hadamard (element-wise) product, and $\mathbf{P}_{intra}$ and $\mathbf{P}_{inter}$ are binary masks identifying edges between nodes with the same and different sensitive attributes.*

Building on Definition 4.1, FairGEM proceeds in three steps. First, the emergence of disparities in graph structural information during generation is analyzed, and an upper bound is established to understand their propagation. Second, it is proven that these structural differences inherently introduce bias into downstream tasks, demonstrating that reducing disparities in structural information within generated graphs improves fairness outcomes. Finally, guided by these theoretical insights, a fairness regularizer is proposed to mitigate differences between intra-group and inter-group edges, effectively addressing graph structural bias.

We begin with the first step of analyzing how graph structural information bias manifests and propagates during the generation process. Theorem 4.2 establishes an upper bound on the graph structural bias that emerges in graph generation (proof in Appendix A).

**Theorem 4.2** *The expected graph structural bias that emerges during the graph generation process, measured by the disparity in adjacency patterns, can be upper bounded by:*

$$\mathbb{E}\|\hat{\mathbf{E}}_0^{dis}\|_F^2 \leq (M^2\|\sigma_.\|_\infty^4 \cdot K)\mathcal{E}(\phi)\left(1 + nK\int_0^1 \Sigma_t^2 \exp\left[nK\int_t^1 \Sigma_z^2 dz\right] dt\right) \tag{5}$$

*where $M$ is a constant determined by various factors (e.g., noise schedule, model architecture, and gradient bounds). Meanwhile, $\Sigma_t^2$ serves as a time-accumulated noise or variance factor, capturing how stochastic perturbations build up over the diffusion process.*

Building on Theorem 4.2, we next analyze how graph structural bias influences the downstream task. Theorem 4.3 demonstrates that by minimizing graph structural bias, we can effectively reduce group disparity in the downstream task (*e.g.*, node classification), with a detailed proof in Appendix B.

**Theorem 4.3** *The structural bias introduced during the graph generation process propagates to downstream tasks, and can be upper bounded by:*

$$\begin{aligned}
\mathbf{h}_D^{(l)} \quad &\leq L\mathbf{M}^{(l-1)}\left[\left\|\mu_{l-1}^{(d)} - \mu_{l-1}^{(f)}\right\|_2 \quad + C\left(\frac{1}{N_d^2}\sum_{p,q\in\mathcal{S}_d}k(\mathbf{h}_p^{(l-1)}, \mathbf{h}_q^{(l-1)}) + \frac{1}{N_f^2}\sum_{r,s\in\mathcal{S}_f}k(\mathbf{h}_r^{(l-1)}, \mathbf{h}_s^{(l-1)})\right.\right. \\
&\left.\left. - \frac{2}{N_d N_f}\sum_{p\in\mathcal{S}_d}\sum_{r\in\mathcal{S}_f}k(\mathbf{h}_p^{(l-1)}, \mathbf{h}_r^{(l-1)})\right)\right] \quad + \left\|\mu^{(d)} - \mu^{(f)}\right\|_2 + L\|\Delta^{(l-1)}\| + C\left\|\Delta_q\right\| + L\sqrt{\mathcal{B}_{spec}(n)}
\end{aligned} \tag{6}$$

*where $L$ is the Lipschitz constant of the activation function, and $C$ is a constant. In addition, $\mu$ denotes the mean representation of a subgroup of nodes.*

Based on Theorem 4.3, we impose an explicit fairness regularizer on the spectral diffusion process to reduce structural bias. Specifically, we combine $\Phi_{struct}$ with the standard score-matching objective to obtain the fair graph structural generation loss:

$$\mathcal{L}_{str} = \mathcal{L}_{score} + \mathbb{E}_t\left[\Phi_{struct}(\Lambda_t)\right] \tag{7}$$

During reverse-time sampling, we modify the drift term of the SDE by adding $\nabla_\Lambda \Phi_{struct}(\bar{\Lambda}_t)$:

$$d\bar{\Lambda}_t = \left( -\frac{1}{2}\sigma_t^2\,\bar{\Lambda}_t - \sigma_t^2\,z_\phi(\bar{\Lambda}_t, t) + \sigma_t^2 \nabla_\Lambda\,\Phi_{\text{struct}}(\bar{\Lambda}_t) \right) d\bar{t} + \sigma_t\,d\bar{\mathbf{W}}_t \tag{8}$$

By doing so, each reverse-time step not only follows the learned score to move toward high-density regions but also corrects for structural bias by minimizing the discrepancy between intra-group and inter-group edges.

## 4.3 Mitigation Feature Bias in Graph Generation Process

FairGEM now proceeds to address feature bias, an important yet largely overlooked aspect in existing fair generative models. Unlike structural bias, however, feature bias demands more nuanced measurement. Specifically, existing methods typically directly measure the differences between all generated node features in deprived and favored groups [43]. While this approach captures the overall differences between subgroups, it ignores the inherent differences of sensitive attributes, leading to a downgrade in generation quality. In other words, fair node feature generation should not erase the inherent differences of sensitive attributes, such as the physiological differences between males and females. To this end, we aim to disentangle node features into sensitive-related features $\mathbf{X}_S$ and sensitive-irrelevant features $\mathbf{X}_{\overline{S}}$. This separation enables a more nuanced approach: minimizing the disparities in the sensitive-irrelevant features ($\mathbf{X}_{\overline{S}}$), while maintaining appropriate differences in the sensitive-related features ($\mathbf{X}_S$), thereby reducing bias while preserving essential group characteristics. We formalize this concept in Definition 4.4.

**Definition 4.4 (Node Feature Generation Bias)** *Given $\mathcal{G}$ with $\mathbf{X}$ and $\mathbf{S}$, we define the node feature generation bias as the distributional discrepancy between subgroups over the sensitive-irrelevant features dimensions. Mathematically, the discrepancy between $\mathbf{X}_{\overline{S}_i}$ of node $v_i$ during the generative process is measured using Maximum Mean Discrepancy (MMD) [44] as follows:*

$$\mathbf{X}_{\overline{S},D} = \frac{1}{|V_{S_d}|^2}\sum_{v_i,v_j \in V_{S_d}} k\left(\mathbf{X}_{\overline{S},i}, \mathbf{X}_{\overline{S},j}\right) + \frac{1}{|V_{S_f}|^2}\sum_{v_i,v_j \in V_{S_f}} k\left(\mathbf{X}_{\overline{S},i}, \mathbf{X}_{\overline{S},j}\right) - \frac{2}{|V_{S_d}|\cdot|V_{S_f}|}\sum_{\substack{v_i \in V_{S_d}\\ v_j \in V_{S_f}}} k\left(\mathbf{X}_{\overline{S},i}, \mathbf{X}_{\overline{S},j}\right) \tag{9}$$

*where $k(\cdot,\cdot)$ is a positive-definite kernel (e.g., RBF). A larger MMD value indicates that the generated distributions of unrelated features differ more between subgroups, implying a higher level of node feature generation bias.*

Building upon this foundation, we introduce a disentanglement-guided diffusion strategy to effectively address biases in node feature generation. Our proposed method employs a generator-refiner framework wherein we initially leverage a Variational Autoencoder (VAE) to separate node features into two distinct components: sensitive-related ($\mathbf{X}_S$) and sensitive-irrelevant ($\mathbf{X}_{\overline{S}}$) attributes. This separation stage helps identify the feature dimensions that should remain unaffected by sensitive information, establishing the groundwork for fair node feature generation. To operationalize this, we introduce two latent variables, $U_S$ and $U_{\overline{S}}$, representing sensitive-related and sensitive-irrelevant information, respectively. Mathematically, we express this probabilistic framework as follows:

$$\begin{aligned} P(S, A, X_S, X_{\overline{S}}, Y) = &\, P(U_S)P(U_{\overline{S}})P(S \mid U_S)P(X_S \mid A, S, U_S)\\ &\, P(A \mid S, U_{\overline{S}}, U_S)P(X_{\overline{S}} \mid A, U_{\overline{S}})P(Y \mid A, U_{\overline{S}}, U_S) \end{aligned} \tag{10}$$

where $P(U_S)$ and $P(U_{\overline{S}})$ denote prior distributions typically modeled as standard normal distributions. The terms $P(X_S \mid A, S, U_S)$ and $P(X_{\overline{S}} \mid A, U_{\overline{S}})$ are responsible for accurately reconstructing sensitive-related and sensitive-irrelated node features, respectively.

To ensure effective disentanglement, we need to enforce independence between the latent variables $U_S$ and $U_{\overline{S}}$. For this purpose, we adopt the Hirschfeld-Gebelein-Rényi (HGR) maximal correlation [45], which generalizes Pearson correlation to capture any non-linear relationship between random variables. Our optimization approach employs an adversarial training mechanism, illustrated in Figure 1. During training, the encoder-decoder parameters are iteratively updated via gradient descent to minimize both the reconstruction loss and the latent dependence, while adversarial networks parameterized by $\omega_{f_1}$ and $\omega_{f_2}$ work in opposition through gradient ascent to detect and amplify remaining latent dependencies. This dynamic interplay of minimization and maximization ensures a

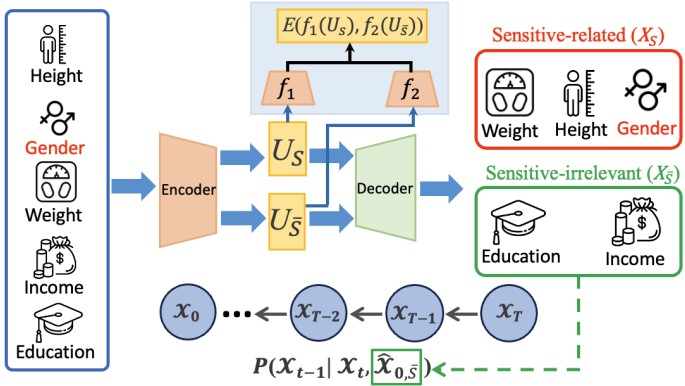

Figure 1: The overview of FairGEM.

thorough and effective disentanglement of sensitive and insensitive feature representations. Finally, we optimize the VAE parameters by maximizing the evidence lower bound (ELBO), formulated as:

$$
\begin{aligned}
\log P(S, A, X_S, X_{\overline{S}}, Y) \geq \mathbb{E}_{q_\phi(U_S, U_{\overline{S}}|S, A, X_S, X_{\overline{S}}, Y)} & \left[ \log P(S \mid U_S) + \log P(A \mid U_S, U_{\overline{S}}, S) \right. \\
& + \log P(X_S \mid U_S, A, S) + \log P(X_{\overline{S}} \mid U_{\overline{S}}, A) + \log P(Y \mid U_S, U_{\overline{S}}, A) \\
& \left. - Q(U_S \mid S, A, X_S) - Q(U_{\overline{S}} \mid S, A, X_{\overline{S}}) + \log P(U_S) + \log P(U_{\overline{S}}) \right] + \lambda \mathcal{L}_D
\end{aligned}
\tag{11}
$$

where the HGR-based penalization term ($\mathcal{L}_D$) is formally defined as: $\mathcal{L}_D = \sup_{f_1, f_2} \frac{\mathbb{E}(f_1(U_S) f_2(U_{\overline{S}}))}{\sqrt{\mathbb{E}(f_1{}^2(U_S)) \mathbb{E}(f_2{}^2(U_{\overline{S}}))}}$, where $f_1$ and $f_2$ are measurable functions with positive and finite variance and $sup(\cdot)$ denotes supremum.

Armed with disentangled $\mathbf{X}_S$ and $\mathbf{X}_{\overline{S}}$, we apply a two-stage generator-refiner approach to ensure fairness in node features generation. The VAE's outputs ($(\hat{\mathbf{X}}_S^{(0)}, \hat{\mathbf{X}}_{\overline{S}}^{(0)})$) serve as the initial signals for a diffusion process that further refines these features. We then enforce the proposed fair node feature generation regularizer to minimize distributional discrepancies between sensitive subgroups in the sensitive-irrelevant features.

As we introduced in Definition 4.4, we formalize our fair node feature generation regularizer as:

$$
\begin{aligned}
\Phi_{\text{feat,ns}}(\mathbf{X}_{\overline{S}, t}) \equiv \mathbf{X}_{\overline{S}, D} = & \frac{1}{|V_{S_0}|^2} \sum_{v_i, v_j \in V_{S_0}} k(\mathbf{X}_{\overline{S}, i, t}, \mathbf{X}_{\overline{S}, j, t}) \\
& + \frac{1}{|V_{S_1}|^2} \sum_{v_i, v_j \in V_{S_1}} k(\mathbf{X}_{\overline{S}, i, t}, \mathbf{X}_{\overline{S}, j, t}) \\
& - \frac{2}{|V_{S_0}||V_{S_1}|} \sum_{\substack{v_i \in V_{S_0} \\ v_j \in V_{S_1}}} k(\mathbf{X}_{\overline{S}, i, t}, \mathbf{X}_{\overline{S}, j, t})
\end{aligned}
\tag{12}
$$

where $\mathbf{X}_{\overline{S}, i, t}$ denotes the sensitive-irrelevant features for node $v_i$ at time $t$.

We integrate this fairness regularizer into a diffusion refiner that evolves the features from time $t$ down to 0. Let $(\mathbf{X}_t)$ represent the set of node feature vectors under noise. The modified reverse-time SDE that incorporates our fairness constraint can be written as:

$$
d\bar{\mathbf{X}}_t = \left( \mathbf{f}^{\mathbf{X}}(\bar{\mathbf{X}}_t, t) - \sigma_{\mathbf{X}, t}^2 z_\theta(\bar{\mathbf{X}}_t, \bar{\mathbf{A}}_t) + \sigma_{\mathbf{X}, t}^2 \nabla_{\mathbf{X}} \Phi_{\text{feat,ns}}(\bar{\mathbf{X}}_{t, \overline{S}}) \right) d\bar{t} + \sigma_{\mathbf{X}, t} d\bar{\mathbf{B}}_t^{\mathbf{X}}
\tag{13}
$$

where $z_\theta(\cdot)$ approximates $\nabla_{\mathbf{X}} \log p(\mathbf{X}_t \mid \hat{\mathbf{X}}_S^{(0)}, \hat{\mathbf{X}}_{\overline{S}}^{(0)})$, and the gradient term $\nabla_{\mathbf{X}} \Phi_{\text{feat,ns}}(\bar{\mathbf{X}}_{t, \overline{S}})$ actively pushes the sensitive-irrelevant features toward smaller cross-group discrepancies.

To learn $z_\theta$ and consistently enforce fairness on $\mathbf{X}_{\overline{S}}$, we augment the standard score-matching objective:

$$\mathcal{L}_{\text{node}}(\theta) = \mathbb{E}_t\left[\left\|z_\theta(\mathbf{X}_t, \mathbf{A}_t) - \nabla \log p_{t|0}(\mathbf{X}_t \mid \mathbf{X}_0)\right\|^2\right] + \xi\,\mathbb{E}_t\left[\Phi_{\text{feat,ns}}(\mathbf{X}_{\overline{S},t})\right] \tag{14}$$

where $\xi$ is a hyperparameter that balances the contribution of the fairness constraint. During reverse diffusion, each step updates node features with both the learned score and the fairness gradient, gradually correcting the unrelated dimensions toward unbiased distributions across subgroups.

In summary, our approach to fair node feature generation proceeds through two complementary stages. First, in the Generator Stage, the VAE disentangles features by sampling latent codes to reconstruct an initial partition $\left(\mathbf{X}_S^{(0)}, \mathbf{X}_{\overline{S}}^{(0)}\right)$, ensuring legitimate group-specific variation in $\mathbf{X}_S$ with minimal entanglement in $\mathbf{X}_{\overline{S}}$. Next, during the Refiner Stage, our diffusion model corrupts $\mathbf{X}^{(0)}$ with noise and learns to reverse this process while imposing the MMD-based fairness penalty on $\mathbf{X}_{\overline{S},t}$.

This designed process yields a final denoised $\mathbf{X}^{(0)}$ that maintains meaningful differences in sensitive features yet achieves unbiased distributions in unrelated features. By leveraging disentangled VAE representations to preserve necessary group distinctions while using a fairness-aware diffusion refiner to align unrelated node features across subgroups, our approach enables achieving fair node feature generation while maintaining better generation quality.

# 5 Experiments

## 5.1 Experiment Setting

**Datasets.** We conduct our experiments using four real-world datasets: **Cora** and **Citeseer** [46]: These widely-used citation network datasets comprise academic papers represented as nodes. Edges indicate citation relationships between papers. Each node's feature vector is generated using a bag-of-words model applied to paper keywords, and the labels denote distinct research areas (topics). **Photo** [47]: This dataset is a subset derived from the Amazon co-purchase network, where nodes represent products and edges represent frequent co-purchasing relationships. Node features consist of bag-of-words vectors extracted from user-generated product reviews. The labels correspond to specific product categories such as cameras, accessories, and lenses. **Pokec-z** [48]: This dataset originates from Pokec, a prominent social networking platform in Slovakia. Nodes represent individual users characterized by detailed attributes including gender, age, geographical location, and personal interests. Edges represent friendship relations among users. The dataset is widely utilized for studying social network dynamics, including community detection and user classification tasks. We use 80% of the graphs in each dataset for training and 20% for testing. Detailed statistical summaries of these datasets are available in Table 1.

Table 1: Summary of the datasets used in the experiments.

| Dataset | # Nodes | # Edges | # Features | Avgrage Degree | Sensitive Attribute |
|---------|---------|---------|------------|----------------|---------------------|
| Cora | 2,708 | 10,556 | 1,433 | 3.89 | Topic |
| Citeseer | 3,327 | 9,228 | 3,703 | 2.77 | Topic |
| Photo | 7,650 | 238,163 | 745 | 31.13 | Product Categories |
| pokec-z | 67,797 | 882,765 | 59 | 10 | Region |

**Baselines.** We compare FairGEM with several state-of-the-art baseline methods across multiple categories: **GRAPHARM** [49]: An autoregressive diffusion-based model that sequentially masks nodes and edges, employing a learned node-ordering strategy for efficient and accurate discrete graph generation. **GSDM** [29]: A framework for graph generation based on spectral diffusion. Using low-rank stochastic differential equations (SDEs) restricted to the space of eigenvalues of the adjacency matrix, the quality of graph topology generation is improved while reducing the computational effort. **FairAdj** [50]: Adjusts the adjacency matrix to achieve dyadic fairness by reducing dependency between link predictions and node sensitive attributes, balancing fairness with predictive accuracy. **FG$^2$AN** [39]: Utilizes adversarial training to jointly optimize node-level and structural fairness, incorporating tailored metrics and strategies to efficiently handle multiple biases

during graph generation. **FairGen** [40]: A deep generative framework that combines label-driven guidance with fairness constraints, leveraging self-paced learning to effectively model protected and unprotected groups from limited labeled data. **FairWire** [6]: Employs diffusion-based techniques with a novel fairness regularizer to mitigate structural bias, effectively preserving fairness in synthetic graph creation without compromising sensitive data. For a fair and consistent comparison, we adapt each baseline method using the original implementations provided by their respective authors. Hyperparameters for these baselines are set according to recommendations from their original papers.

**Evaluation Metrics.** We evaluated FairGEM in two perspectives: **i) Quality of Generated Graphs:** Building on [49], we employ Maximum Mean Discrepancy (MMD) to compare generated and original graphs in terms of degree distributions (DD), clustering coefficients (Clus), and node features (NFea), with smaller MMD signifying closer fidelity. To further evaluate structural fairness, we introduce three metrics: Fair Degree Distribution (Fair-DD), Fair Clustering Coefficient (Fair-Clus), and Fair Node Feature (Fair-NFea), each capturing cross-subgroup disparities. These metrics take the form $f(\mathcal{G}_{S_0}, \tilde{\mathcal{G}}_{S_0}) - f(\mathcal{G}_{S_1}, \tilde{\mathcal{G}}_{S_1})$, where $\mathcal{G}_{S_i}$ and $\tilde{\mathcal{G}}_{S_i}$ refer to the induced subgraphs of the real and generated data on subgroup $S_i$, and $f(\cdot)$ denotes the MMD calculated function, with small value reflecting a fairer outcomes. **ii) Node Classification Performance:** We measure the utility in node classification tasks using Accuracy and F1 scores, while quantifying the fairness of these results with $\Delta$DP [51] and $\Delta$EO [52], where smaller values indicate fairer outcomes.

## 5.2 Experiment Results

**Quality of generated graphs.** Table 2 summarizes the generation performance of all methods across each dataset, evaluating models in terms of both quality and fairness, with additional results included in Appendix C.2. As the results indicate, FairGEM demonstrates competitive or superior generation quality compared to the baseline approaches, consistently showing smaller discrepancies in key graph statistics such as degree distributions and clustering coefficients. At the same time, it substantially improves fairness metrics, suggesting that its generated node features and structural patterns do not disproportionately favor any subgroup. The strong performance can be attributed to two key factors. i) By disentangling node features into sensitive-related and sensitive-unrelated components, FairGEM preserves meaningful group-specific differences without introducing unintended biases. ii) FairGEM incorporates an explicit fairness regularizer within the generative diffusion process, actively penalizing biases in sensitive subpopulations. Hence, FairGEM not only produces realistic and coherent graph samples but also ensures equitable treatment of different groups.

Table 2: Graph generation results on Cora and Pokec-z datasets.

| Method | Cora | | | | | | Pokec-z | | | | | |
| --- | --- | --- | --- | --- | --- | --- | --- | --- | --- | --- | --- | --- |
| | DD | Clus | NFea | Fair-DD | Fair-Clus | Fair-NFea | DD | Clus | NFea | Fair-DD | Fair-Clus | Fair-NFea |
| GRAPHARM | **0.238** | 0.135 | 0.312 | 0.077 | 0.083 | 0.049 | 0.348 | 0.150 | 0.253 | 0.078 | 0.067 | 0.038 |
| GSDM | 0.241 | **0.128** | **0.289** | 0.064 | 0.069 | 0.051 | **0.331** | **0.142** | **0.231** | 0.073 | 0.060 | 0.031 |
| FairAdj | 0.258 | 0.157 | 0.321 | 0.035 | 0.043 | 0.032 | 0.358 | 0.168 | 0.281 | 0.055 | 0.039 | 0.023 |
| FG$^2$AN | 0.263 | 0.169 | 0.335 | 0.038 | 0.047 | 0.036 | 0.374 | 0.178 | 0.295 | 0.059 | 0.046 | 0.028 |
| FairGen | 0.275 | 0.173 | 0.348 | 0.027 | 0.045 | 0.031 | 0.383 | 0.185 | 0.311 | 0.053 | 0.038 | 0.037 |
| FairWire | 0.259 | 0.161 | 0.332 | 0.031 | 0.043 | 0.038 | 0.370 | 0.170 | 0.281 | 0.045 | 0.032 | 0.034 |
| **FairGEM** | 0.233 | 0.142 | 0.307 | **0.020** | **0.035** | **0.019** | 0.357 | 0.161 | 0.250 | **0.040** | **0.023** | **0.017** |

**Downstream task performance evaluation.** To evaluate the impact of our graph generative model on downstream tasks, we evaluated its performance and fairness on node classification using generated graphs. We conducted experiments on four datasets, with detailed results for Cora and Citeseer presented in Table 3 and the remaining results included in the Appendix C.2 due to space constraints. For each dataset, we trained a standard GCN model [53] on graphs generated by different methods and assessed both accuracy and fairness metrics. All experiments were repeated five times with the average results reported. The results demonstrate that graphs generated by FairGEM consistently enhance both the accuracy and fairness of node classification outcomes compared to other graph generation model baselines. For instance, on the Cora dataset, FairGEM generated graphs achieved a 21.2% improvement in $\Delta_{DP}$, while maintaining comparable accuracy to the original graph. This improvement can be attributed to FairGEM's comprehensive approach in mitigating both graph structural and feature biases during the graph generation process, effectively limiting the propagation of these biases into downstream tasks.

Table 3: Node classification results on Cora and Pokec-z datasets.

| Method | Cora | | | | Pokec-z | | | |
|---|---|---|---|---|---|---|---|---|
| | Acc (%) | F1-score (%) | $\Delta_{DP}$ (%) | $\Delta_{EO}$ (%) | Acc (%) | F1-score (%) | $\Delta_{DP}$ (%) | $\Delta_{EO}$ (%) |
| Original-GCN | **82.43 ± 0.34** | **84.40 ± 3.60** | 27.01 ± 1.38 | 25.21 ± 1.13 | 76.31 ± 1.34 | **68.47 ± 1.28** | 20.11 ± 1.67 | 22.31 ± 0.98 |
| GRAPHARM-GCN | 81.03 ± 0.23 | 82.70 ± 2.31 | 25.21 ± 1.38 | 21.31 ± 1.43 | 73.25 ± 2.01 | 65.21 ± 1.61 | 16.98 ± 1.21 | 18.64 ± 1.38 |
| GSDM-GCN | 81.51 ± 1.23 | 83.91 ± 0.95 | 25.47 ± 1.24 | 23.78 ± 0.77 | **76.88 ± 0.79** | 67.71 ± 1.59 | 19.58 ± 2.16 | 20.31 ± 1.15 |
| FairAdj-GCN | 77.77 ± 1.64 | 78.32 ± 1.88 | 17.13 ± 6.36 | 13.96 ± 2.24 | 70.93 ± 1.59 | 60.32 ± 1.23 | 14.25 ± 1.51 | 15.48 ± 1.09 |
| FG$^2$AN-GCN | 78.10 ± 0.81 | 78.88 ± 1.72 | 18.66 ± 4.30 | 14.05 ± 0.32 | 71.82 ± 1.79 | 59.48 ± 0.98 | 15.16 ± 1.27 | 16.35 ± 1.90 |
| FairGen-GCN | 79.54 ± 1.56 | 80.54 ± 2.16 | 14.16 ± 0.89 | 13.35 ± 1.24 | 73.43 ± 1.77 | 63.71 ± 1.87 | 12.11 ± 1.19 | 12.81 ± 1.54 |
| FairWire-GCN | 78.21 ± 1.03 | 79.67 ± 1.55 | 14.76 ± 0.24 | 13.65 ± 0.51 | 74.98 ± 1.01 | 61.11 ± 1.09 | 12.98 ± 1.36 | 14.01 ± 2.10 |
| **FairGEM-GCN** | 79.75 ± 0.98 | 80.36 ± 1.16 | **11.71 ± 1.24** | **10.15 ± 1.07** | 75.25 ± 1.81 | 65.39 ± 1.03 | **9.23 ± 0.59** | **11.46 ± 1.12** |

**Ablation study.** We conduct ablation studies to gain insights into the effect of each fairness regularizer in FairGEM on improving fairness and graph generation quality. Specifically, we create three ablated versions: 1) FairGEM-WS removes the fair graph structure regularizer, 2) FairGEM-WF removes the fair node feature regularizer, and 3) FairGEM-WD removes the disentanglement component and directly applies fairness constraints to all node features indiscriminately. Figure 3 presents ablation results across the Cora and Pokec-z datasets, with additional results included in Appendix C.2, revealing several key findings. First, compared to FairGEM and FairGEM-WS, FairGEM-WD shows decreased generation quality because applying fairness constraints to all node features leads to unrealistic consistency, thereby reducing generation quality. Second, FairGEM-WS exhibits stronger intra-group connectivity compared to other models, highlighting the critical role of our fair graph structure regularizer in reducing graph structural bias. Third, FairGEM-WD shows only slightly improved node feature fairness compared to FairGEM-WF, indicating that forcing consistency across all features may introduce additional bias that counteracts fairness improvements. Finally, the full FairGEM model consistently outperforms all ablated versions in terms of fairness metrics, validating the necessity and complementarity of each component in our design.

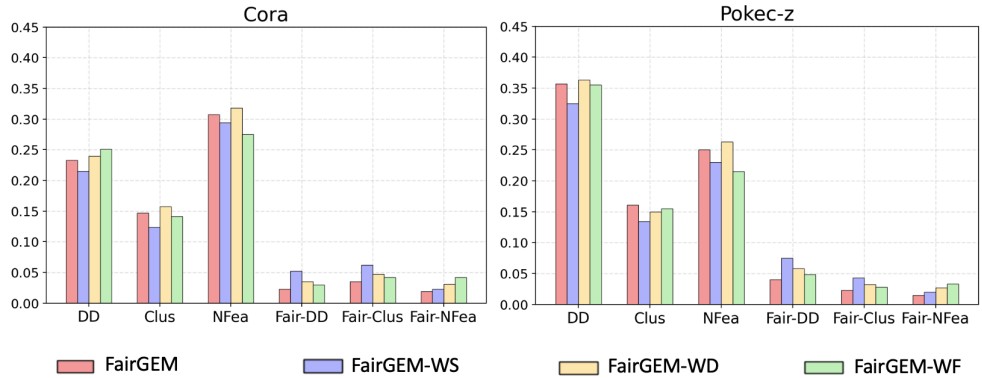

Figure 2: Ablation study results for FairGEM, FairGEM-WS, FairGEM-WD and FairGEM-WF in Cora and Pokec-z datasets.

# 6 Conclusion

Generating synthetic graphs that reflect real-world structural properties has emerged as a promising solution for scalability and privacy needs in real-world networks. However, fairness remains largely unexplored in graph generation models. To bridge this gap, this paper proposes FairGEM, a one-shot generative framework designed to mitigate both structural and feature-level biases. By departing from the autoregressive model that suffers from ordering sensitivities, FairGEM transforms random noise directly into a final, bias-corrected graph, avoiding the pitfalls of node or edge ordering dependencies. FairGEM incorporates a theoretically grounded fairness regularizer into the diffusion process, effectively identifying and reducing real bias factors. Comprehensive experiments on real datasets confirm that FairGEM outperforms state-of-the-art baselines, offering superior bias mitigation without compromising generative quality. These results establish a solid foundation for future work on one-shot fair graph generation.

## Acknowledgements

This work was supported in part by the National Science Foundation (NSF) under Grant No. 2404039.

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

# A   Proof of Theorem 4.2

Without loss of generality, we will focus on providing the reconstruction disparity bounds for adjacency matrix generation. Consider a fairness-aware spectral diffusion model reconstructing an adjacency matrix $\hat{\mathbf{A}}$ from the original graph $\mathcal{G}$ with adjacency matrix $\mathbf{A}$. Let $\mathbf{P}_{\text{intra}}$ and $\mathbf{P}_{\text{inter}}$ denote binary masks for intra-group and inter-group edges respectively. We define the disparity between intra- and inter-group as follows:

$$\mathbf{E}_t^{\text{dis}} \triangleq \mathbf{E}_t^{\text{inter}} - \mathbf{E}_t^{\text{intra}} \tag{15}$$

where $\mathbf{E}_t \triangleq \hat{\mathbf{A}}_t - \mathbf{A}$ denotes the reconstruction error. In addition, the term $\mathbf{E}_t^{\text{inter}} \triangleq \mathbf{P}_{\text{inter}} \odot \mathbf{E}_t$ and $\mathbf{E}_t^{\text{intra}} \triangleq \mathbf{P}_{\text{intra}} \odot \mathbf{E}_t$.

Building on this, we can defined the disparity between intra- and inter-group at time $t$ as follows:

$$d\mathbf{E}_t^{\text{dis}} = (\mathbf{P}_{\text{inter}} - \mathbf{P}_{\text{intra}}) \odot d\hat{\mathbf{A}}_t = d\hat{\mathbf{A}}_t^{\text{inter}} - d\hat{\mathbf{A}}_t^{\text{intra}} \tag{16}$$

Hence, we have:

$$d\bar{\Lambda}_t = \left( -\frac{1}{2}\sigma_t^2 \bar{\Lambda}_t - \sigma_t^2 z_\phi(\bar{\Lambda}_t, t) + \sigma_t^2 \nabla_\Lambda \Phi_{\text{struct}}(\bar{\Lambda}_t) \right) d\bar{t} + \sigma_t d\bar{\mathbf{W}}_t. \tag{17}$$

Given the spectral decomposition $\mathbf{A} = \mathbf{U}\Lambda\mathbf{U}^\top$, the reverse-time spectral diffusion SDE for $\hat{\mathbf{A}}_t$ with fairness regularization $\Phi_{\text{struct}}(\Lambda)$ is:

$$d\hat{\mathbf{A}}_t = \left( -\frac{1}{2}\sigma_t^2 \hat{\mathbf{A}}_t - \sigma_t^2 z_\phi(\hat{\mathbf{A}}_t, t) + \sigma_t^2 \mathbf{U}\nabla_\Lambda \Phi_{\text{struct}}(\Lambda_t)\mathbf{U}^\top \right) d\bar{t} + \sigma_t d\bar{\mathbf{M}}_t. \tag{18}$$

Therefore, the $d\hat{\mathbf{A}}_t^{\text{inter}}$ and $d\hat{\mathbf{A}}_t^{\text{intra}}$ are:

$$\begin{cases} d\hat{\mathbf{A}}_t^{\text{inter}} = \left( -\frac{1}{2}\sigma_t^2 \hat{\mathbf{A}}_t^{\text{inter}} - \sigma_t^2 \mathbf{P}_{\text{inter}} \odot z_\phi(\hat{\mathbf{A}}_t, t) + \sigma_t^2 \mathbf{P}_{\text{inter}} \odot \left[ \mathbf{U}\nabla_\Lambda \Phi_{\text{struct}} \mathbf{U}^\top \right] \right) d\bar{t} + \sigma_t \mathbf{P}_{\text{inter}} \odot d\bar{\mathbf{M}}_t \\ d\hat{\mathbf{A}}_t^{\text{intra}} = \left( -\frac{1}{2}\sigma_t^2 \hat{\mathbf{A}}_t^{\text{intra}} - \sigma_t^2 \mathbf{P}_{\text{intra}} \odot z_\phi(\hat{\mathbf{A}}_t, t) + \sigma_t^2 \mathbf{P}_{\text{intra}} \odot \left[ \mathbf{U}\nabla_\Lambda \Phi_{\text{struct}} \mathbf{U}^\top \right] \right) d\bar{t} + \sigma_t \mathbf{P}_{\text{intra}} \odot d\bar{\mathbf{M}}_t \end{cases} \tag{19}$$

To bound disparity, observe that the supports of $\mathbf{P}_{\text{inter}}$ and $\mathbf{P}_{\text{intra}}$ are disjoint, hence:

$$\|\mathbf{E}_t^{\text{dis}}\|_F^2 = \|\mathbf{E}_t^{\text{inter}} - \mathbf{E}_t^{\text{intra}}\|_F^2 \leq \|\mathbf{E}_t^{\text{inter}}\|_F^2 + \|\mathbf{E}_t^{\text{intra}}\|_F^2 = \|(\mathbf{P}_{\text{inter}} + \mathbf{P}_{\text{intra}}) \odot \mathbf{E}_t\|_F^2 \leq \|\mathbf{E}_t\|_F^2 \tag{20}$$

Taking expectation at $t = 0$ and applying the spectral-noise variant gives. Hence, the disparity between the two subgroups satisfies:

$$\mathbb{E}\|\mathbf{E}_0^{\text{dis}}\|_F^2 \leq \mathbb{E}\|\mathbf{E}_0\|_F^2 \leq \mathcal{B}_{\text{spec}}(n) \tag{21}$$

Building on this, the final disparity between the inter- and intra- edges is:

$$\mathbb{E}\|\mathbf{E}_0^{\text{dis}}\|_F^2 \leq \mathcal{B}_{\text{spec}}(n) = \left( M^2 \|\sigma_\cdot\|_\infty^4 \cdot K \right) \mathcal{E}(\phi) \left( 1 + nK \int_0^1 \Sigma_t^{-2} \exp\left[ nK \int_t^1 \Sigma_z^{-2} \, dz \right] dt \right). \tag{22}$$

where $K \triangleq 2ML/\mathbb{E}\|\mathbf{A}\|_{2,2}$, $M, L$ are absolute constants, $\Sigma_t^2 \triangleq 1 - \exp\left( -\int_0^t \sigma_z^2 \, dz \right)$, and $\mathcal{E}(\cdot)$ is the expected score and defined as:

$$\mathcal{E}(\cdot) \triangleq \mathbb{E}_{\mathbf{z}\sim\mathcal{D}} \mathbb{E}_{\mathbf{z}_t|\mathbf{z}} \|z_\theta(\mathbf{z}_t) - \nabla \log p_t(\mathbf{z}_t)\|^2 \tag{23}$$

This result demonstrates that the fairness-aware spectral diffusion model effectively controls reconstruction disparity between intra- and inter-group connections. The disparity measure is guaranteed to be bounded in terms of the spectral reconstruction bound $\mathcal{B}_{\text{spec}}(n)$, indicating controlled fairness in graph structure generation.

This completes the proof.

## B  Proof of Theorem 4.3

For binary node classification, the group disparity such as statistical group parity is defined as: $\Delta_{SP} = |P(\hat{y} = 1|s = d) - P(\hat{y} = 1|s = f)|$. To analyze how structural bias propagates to this fairness metric, we examine the properties of the Softmax function, which generates the prediction probabilities $P_1$ and $P_2$ for classes $c_1$ and $c_2$, respectively. A key analytical property of Softmax is its Lipschitz continuity with constant $L$, which guarantees that differences in output probabilities are bounded proportionally to differences in input vectors.

$$\|f(\mathbf{z}_i) - f(\mathbf{z}_j)\|_1 \ \leq \ L \|\mathbf{z}_i - \mathbf{z}_j\|_2 \ \leq \ L \|W^l\| \|\mathbf{h}_i^l - \mathbf{h}_j^l\|_2 \tag{24}$$

where $\mathbf{z}_i = W^l \mathbf{h}_i^l$, and $\mathbf{z}_j = W^l \mathbf{h}_j^l$. In this formulation, $\mathbf{h}_i^l$ represents the node embedding for some $v_i \in S_d$ and $\mathbf{h}_j^l$ represents the node embedding for some $v_j \in S_f$, where $S_d$ and $S_f$ are the two sensitive groups. Without loss of generality, we focus on the $l^{th}$ GNN layer to illustrate this propagation mechanism, where the input representations are denoted by $\mathbf{h}^l$ and the corresponding output representations are denoted by $\mathbf{h}^{l+1}$.

Hence, we can rewrite the statistical parity as follows:

$$\Delta_{\text{DP}} = \left| \frac{1}{N_0} \sum_{i \in \mathcal{S}_0} f(\mathbf{z}_i)_1 \ - \ \frac{1}{N_1} \sum_{j \in \mathcal{S}_1} f(\mathbf{z}_j)_1 \right| \tag{25}$$

Building on this, we can conclude that disparities in node classification outcomes directly stem from discrepancies in node representations. To quantify node representation discrepancies ($\mathbf{h}_{\overline{S},D}^l$) on the $l^{th}$ GNN layer, we adopt Maximum Mean Discrepancy (MMD) [44] as our measurement framework. MMD offers superior transfer learning capabilities compared to alternative metrics [54, 55, 56], making it particularly well-suited for addressing fairness challenges in graph learning contexts. Hence, the node representation discrepancies are defined as:

$$
\begin{aligned}
\mathbf{h}_{\overline{S},D}^l &= \text{MMD}\Big( \{\mathbf{h}_{S,i}^l \mid v_i \in V_{S_d}\}, \ \{\mathbf{h}_{S,i}^l \mid v_i \in V_{S_f}\} \Big) \\
&= \frac{1}{|V_{S_d}|^2} \sum_{v_i, v_j \in V_{S_d}} k\Big( \mathbf{h}_{\overline{S},i}^l, \ \mathbf{h}_{\overline{S},j}^l \Big) + \frac{1}{|V_{S_f}|^2} \sum_{v_i, v_j \in V_{S_f}} k\Big( \mathbf{h}_{\overline{S},i}^l, \ \mathbf{h}_{\overline{S},j}^l \Big) \\
&\quad - \frac{2}{|V_{S_d}| \cdot |V_{S_f}|} \sum_{\substack{v_i \in V_{S_d} \\ v_j \in V_{S_f}}} k\Big( \mathbf{h}_{\overline{S},i}^l, \ \mathbf{h}_{\overline{S},j}^l \Big)
\end{aligned} \tag{26}
$$

where $k(x, y) = \exp\left( -\gamma \|x - y\|^2 \right)$ is RBF kernel function.

Building on this, and given that Graph Attention Networks [57] (GAT) adopt the message passing by assigning different weights to neighbor nodes as:

$$\mathbf{h}_i^{(l)} = \sum_{v_j \in \mathcal{N}(i)} a_{ij}^{(l-1)} \mathbf{h}_j^{(l-1)} \quad \text{with} \quad a_{ij}^{(l-1)} = \frac{\exp\left( e_{ij}^{(l-1)} \right)}{\sum_{v_j \in \mathcal{N}(i)} \exp\left( e_{ij}^{(l-1)} \right)}, \tag{27}$$

Here, we further distinguish between neighbor information from the inter-group and neighbor information from the intra-group, as detailed as follows:

$$\mathbf{h}_i^{(l)} = \sum_{v_j \in \mathcal{N}_{\text{intra}}(i)} \alpha_{ij,\text{intra}}^{(l-1)} \mathbf{h}_j^{(l-1)} + \sum_{v_j \in \mathcal{N}_{\text{inter}}(i)} \alpha_{ij,\text{inter}}^{(l-1)} \mathbf{h}_j^{(l-1)}. \tag{28}$$

where $\alpha_{ij,\text{intra}}^{(l-1)}$ and $\alpha_{ij,\text{inter}}^{(l-1)}$ are defined as:

$$\alpha_{ij,\text{intra}}^{(l-1)} = \frac{\exp\left(e_{ij,\text{intra}}^{(l-1)}\right)}{\sum\limits_{v_k \in \mathcal{N}_{\text{intra}}(i)} \exp\left(e_{ik,\text{intra}}^{(l-1)}\right)}, \quad \alpha_{ij,\text{inter}}^{(l-1)} = \frac{\exp\left(e_{ij,\text{inter}}^{(l-1)}\right)}{\sum\limits_{v_k \in \mathcal{N}_{\text{inter}}(i)} \exp\left(e_{ik,\text{inter}}^{(l-1)}\right)}. \tag{29}$$

Hence, we can rewrite the node representation as follows:

$$
\begin{aligned}
\mathbf{h}_i^{(l)} &= \mathbf{h}_i^{(l-1)} + \sum_{j \in \mathcal{N}(i)} a_{ij}^{(l-1)} \mathbf{h}_j^{(l-1)} - \Delta_{bias}^{(l-1)} \\
&= \mathbf{h}_i^{(l-1)} + w_{i,\text{intra}}^{(l-1)} \sum_{j \in \mathcal{N}_{\text{intra}}(i)} a_{ij,\text{intra}}^{(l-1)} \mathbf{h}_j^{(l-1)} \\
&\quad + w_{i,\text{inter}}^{(l-1)} \sum_{j \in \mathcal{N}_{\text{inter}}(i)} a_{ij,\text{inter}}^{(l-1)} \mathbf{h}_j^{(l-1)} - LC\left[ \frac{1}{N_d^2} \sum_{u \in S_d} k(\mathbf{h}_i^{(l-1)}, \mathbf{h}_u^{(l-1)})(\mathbf{h}_i^{(l-1)} - \mathbf{h}_u^{(l-1)}) \right. \\
&\quad \left. - \frac{1}{N_d N_f} \sum_{v \in S_f} k(\mathbf{h}_i^{(l-1)}, \mathbf{h}_v^{(l-1)})(\mathbf{h}_i^{(l-1)} - \mathbf{h}_v^{(l-1)}) \right]
\end{aligned}
\tag{30}
$$

where $\mathbf{h}_u^{(l-1)} = \left( \alpha_{iu,\text{intra}}^{(l-1)} \mathbf{h}_{u,\text{intra}}^{(l-1)} + \alpha_{iu,\text{inter}}^{(l-1)} \mathbf{h}_{u,\text{inter}}^{(l-1)} \right)$ and similarly for others.

For nodes belonging to the sensitive group $S_d$, the representation $h_u^{(l)}$ at layer $l$ is constrained within a hypercube centered at the group mean $\mu_l^{(d)}$ with boundaries defined by deviation vector $\Delta^l$, expressed as $\mu_l^{(d)} - \Delta^l \preceq h_u^{(l)} \preceq \mu_l^{(d)} + \Delta^l$ [58]. This constraint implies that each dimension $m$ of the representation vector exists within a specific interval $[\mu_{l,m}^{(d)} - \Delta_m^l, \mu_{l,m}^{(d)} + \Delta_m^l]$. Analogously, representations of nodes from group $S_f$ are bounded within their own characteristic region $[\mu_l^{(f)} \pm \Delta^l]$.

$$
\begin{aligned}
\mathbf{h}_i^{(l)} \in &\left[ \mu_{l-1}^{(d)} + w_{i,\text{intra}}^{(l-1)} \sum_{u \in \mathcal{N}_{\text{intra}}(i)} a_{iu,\text{intra}}^{(l-1)} \mathbf{h}_u^{(l-1)} + w_{i,\text{inter}}^{(l-1)} \sum_{u \in \mathcal{N}_{\text{inter}}(i)} a_{iu,\text{inter}}^{(l-1)} \mathbf{h}_u^{(l-1)} \right. \tag{31} \\
&\quad - LC\left( \frac{1}{N_d^2} \sum_{u \in S_d} k\big(\mathbf{h}_i^{(l-1)}, \mathbf{h}_u^{(l-1)}\big)\big(\mathbf{h}_i^{(l-1)} - \mathbf{h}_u^{(l-1)}\big) - \right. \\
&\quad \left. \frac{1}{N_d N_f} \sum_{v \in S_f} k\big(\mathbf{h}_i^{(l-1)}, \mathbf{h}_v^{(l-1)}\big)\big(\mathbf{h}_i^{(l-1)} - \mathbf{h}_v^{(l-1)}\big) \right) \tag{32} \\
&\quad \left. \pm \left[ L\,\boldsymbol{\Delta}^{(l-1)} + 2\sqrt{N}\,\Delta_q \right] \right]
\end{aligned}
$$

Therefore, the node representation discrepancy is:

$$\left\|\frac{1}{N_d}\sum_{i\in\mathcal{S}_d}\mathbf{h}_i^{(l)} - \frac{1}{N_f}\sum_{j\in\mathcal{S}_f}\mathbf{h}_j^{(l)}\right\|_2 \leq \left(1 - \frac{1}{N_d}\sum_{i\in\mathcal{S}_d}\beta_i^{(l-1)} - \frac{1}{N_f}\sum_{j\in\mathcal{S}_f}\beta_j^{(l-1)}\right)\|\mu_{l-1}^{(d)} - \mu_{l-1}^{(f)}\|_2$$

$$+ C\left(\frac{1}{N_dN_f^2} + \frac{1}{N_d^2N_f}\right)\left[\frac{1}{N_d^2}\sum_{p,q\in\mathcal{S}_d}k(\mathbf{h}_p^{(l-1)}, \mathbf{h}_q^{(l-1)}) + \frac{1}{N_f^2}\sum_{r,s\in\mathcal{S}_f}k(\mathbf{h}_r^{(l-1)}, \mathbf{h}_s^{(l-1)})\right.$$

$$\left. - \frac{2}{N_dN_f}\sum_{p\in\mathcal{S}_d}\sum_{r\in\mathcal{S}_f}k(\mathbf{h}_p^{(l-1)}, \mathbf{h}_r^{(l-1)})\right] + L\left\|\Delta^{(l-1)}\right\| + 2\sqrt{N}\,\Delta_q \tag{33}$$

Building on this, we define the upper bound of the consequent node representation discrepancy on node representation between two sensitive subgroups as follows:

$$\mathbf{h}_D^{(l)} = \left\|\frac{1}{N_d}\sum_{i\in\mathcal{S}_d}\mathbf{h}_i^{(l)} - \frac{1}{N_f}\sum_{j\in\mathcal{S}_f}\mathbf{h}_j^{(l)}\right\|_2$$

$$\leq \left(1 - \frac{1}{N_d}\sum_{i\in\mathcal{S}_d}\beta_i^{(l-1)} - \frac{1}{N_f}\sum_{j\in\mathcal{S}_f}\beta_j^{(l-1)}\right)\|\mu_{l-1}^{(d)} - \mu_{l-1}^{(f)}\|_2$$

$$+ C\left(\frac{1}{N_dN_f^2} + \frac{1}{N_d^2N_f}\right)\left[\frac{1}{N_d^2}\sum_{p,q\in\mathcal{S}_d}k(\mathbf{h}_p^{(l-1)}, \mathbf{h}_q^{(l-1)}) + \frac{1}{N_f^2}\sum_{r,s\in\mathcal{S}_f}k(\mathbf{h}_r^{(l-1)}, \mathbf{h}_s^{(l-1)})\right.$$

$$\left. - \frac{2}{N_dN_f}\sum_{p\in\mathcal{S}_d}\sum_{r\in\mathcal{S}_f}k(\mathbf{h}_p^{(l-1)}, \mathbf{h}_r^{(l-1)})\right] + \|\mu^{(d)} - \mu^{(f)}\|_2 + L\left\|\Delta^{(l-1)}\right\| + C\left\|\Delta_q\right\| \tag{34}$$

Building on this theoretical foundation, we analyze how graph generation models introduce disparity between node representations, *i.e.*, graph structure information generation bias. Mathematically, this bias can be represented as:

$$\Delta_{\text{gen}}^{(l)} = \left\|\mu_{l,\text{gen}}^{(d)} - \mu_{l,\text{gen}}^{(f)}\right\| - \left\|\mu_l^{(d)} - \mu_l^{(f)}\right\| \tag{35}$$

Based on the GNN layer's aggregation and activation functions having bounded Lipschitz constants with respect to the inputs, then any changes in the adjacency matrix propagate through the network in a controlled way, *i.e.*, the discrepancy in final-layer node representations is also bounded by a proportional factor [58].

$$\Delta_{\text{gen}}^{(l)} \leq L\sqrt{\mathbb{E}\|E_0\|_F^2} \leq L\sqrt{\mathcal{B}_{\text{spec}}(n)} \tag{36}$$

Given that the graph structure information generation bias between the inter- and intra- edges in Equation 22. Therefore, the final result for node representation discrepancy can be bounded by:

$$\mathbf{h}_D^{(l)} = \left\| \frac{1}{N_d} \sum_{i \in \mathcal{S}_d} \mathbf{h}_i^{(l)} - \frac{1}{N_1} \sum_{j \in \mathcal{S}_1} \mathbf{h}_j^{(l)} \right\|_2$$

$$\leq L\mathbf{M}^{(l-1)} \Bigg[ \left\| \mu_{l-1}^{(d)} - \mu_{l-1}^{(f)} \right\|_2$$

$$+ C \left( \frac{1}{N_d^2} \sum_{p,q \in \mathcal{S}_d} k(\mathbf{h}_p^{(l-1)}, \mathbf{h}_q^{(l-1)}) + \frac{1}{N_f^2} \sum_{r,s \in \mathcal{S}_f} k(\mathbf{h}_r^{(l-1)}, \mathbf{h}_s^{(l-1)}) \right.$$

$$\left. \left. - \frac{2}{N_d N_f} \sum_{p \in \mathcal{S}_d} \sum_{r \in \mathcal{S}_f} k(\mathbf{h}_p^{(l-1)}, \mathbf{h}_r^{(l-1)}) \right) \right]$$

$$+ \left\| \mu^{(d)} - \mu^{(f)} \right\|_2 + L\|\Delta^{(l-1)}\| + C\|\Delta_q\| + L\sqrt{\mathcal{B}_{\mathrm{spec}}(n)} \tag{37}$$

which concludes the proof.

## C  Experiments

### C.1  Implementation Details

Models are trained for {500, 500, 500, 200} epochs on Cora, Citeseer, Photo, and Pokec, respectively, using the Adam optimizer with betas = (0.9, 0.999), learning rate $= 1 \times 10^{-3}$, and weight decay $= 1 \times 10^{-5}$. For node features $\mathbf{X}$, adjacency matrix $\mathbf{A}$, and latent codes $\mathbf{u}$, we adopt identical variance-preserving stochastic differential equations with $\beta_{\min} = 0.1$, $\beta_{\max} = 1.0$, and 1000 discrete time steps. During inference, we start from standard Gaussian noise, run the predictor–corrector chain for all 1000 steps, include a final deterministic noise-removal step, and stop at $\varepsilon = 1 \times 10^{-4}$. We use mini-batches of size 32, early stopping with a patience of 30 epochs, and retain weights from the best epoch. Our VAE architecture consists of GCN layers with ReLU activation for encoding and decoding, and the discriminator employs fully connected layers with LeakyReLU activation. For the downstream GCN task, we use a 1-layer GCN with 16 hidden units and a linear classifier. All experiments are implemented in PyTorch.

### C.2  Additional Experimental Results

**Additional results for quality of generated graphs.** Table 4 presents additional results on the Photo and Citeseer datasets. The results demonstrating that FairGEM consistently achieves competitive or superior generation quality compared to baseline methods. Specifically, FairGEM maintains smaller discrepancies in important graph statistics, such as degree distributions and clustering coefficients, across both datasets. Furthermore, it continues to exhibit significant improvements in fairness metrics, indicating that the generated node features and structural patterns effectively avoid disproportionate favoring of any subgroup. This consistently strong performance further supports the effectiveness of FairGEM's approach, emphasizing its ability to generate realistic, and unbiased synthetic graph.

Table 4: Graph generation results on Photo and Citeseer datasets.

| Method | Photo | | | | | | Citeseer | | | | | |
|---|---|---|---|---|---|---|---|---|---|---|---|---|
| | DD | Clus | NFea | Fair-DD | Fair-Clus | Fair-NFea | DD | Clus | NFea | Fair-DD | Fair-Clus | Fair-NFea |
| GRAPHARM | 0.317 | 0.235 | 0.327 | 0.063 | 0.084 | 0.068 | **0.265** | 0.193 | 0.163 | 0.098 | 0.054 | 0.058 |
| GSDM | **0.293** | **0.229** | **0.314** | 0.055 | 0.079 | 0.063 | 0.271 | **0.187** | **0.152** | 0.084 | 0.047 | 0.042 |
| FairAdj | 0.301 | 0.231 | 0.320 | 0.036 | 0.055 | 0.051 | 0.337 | 0.204 | 0.158 | 0.067 | 0.026 | 0.035 |
| FG$^2$AN | 0.357 | 0.253 | 0.345 | 0.042 | 0.060 | 0.050 | 0.358 | 0.221 | 0.167 | 0.071 | 0.035 | 0.051 |
| FairGen | 0.378 | 0.294 | 0.368 | 0.038 | 0.058 | 0.041 | 0.377 | 0.237 | 0.181 | 0.068 | 0.025 | 0.036 |
| FairWire | 0.347 | 0.287 | 0.354 | 0.029 | 0.048 | 0.049 | 0.349 | 0.218 | 0.173 | 0.062 | 0.023 | 0.039 |
| **FairGEM** | 0.318 | 0.245 | 0.331 | **0.018** | **0.033** | **0.037** | 0.311 | 0.187 | 0.160 | **0.051** | **0.013** | **0.028** |

**Additional results for downstream task performance evaluation.** Table 5 presents these supplementary results. Specifically, across these datasets, synthetic graphs generated by FairGEM consistently led to improved fairness in node classification tasks when compared to baseline generation methods. These consistent improvements underline FairGEM's effectiveness in limiting bias propagation from generated graphs into downstream applications, thereby enhancing fairness in node classification task.

Table 5: Node classification results on Photo and Citeseer datasets.

| Method | Photo | | | | Citeseer | | | |
|---|---|---|---|---|---|---|---|---|
| | Acc (%) | F1-score (%) | $\Delta_{DP}$ (%) | $\Delta_{EO}$ (%) | Acc (%) | F1-score (%) | $\Delta_{DP}$ (%) | $\Delta_{EO}$ (%) |
| Original-GCN | **74.83 ± 2.18** | **82.36 ± 1.84** | 13.21 ± 0.82 | 14.54 ± 1.56 | 76.31 ± 1.34 | **68.47 ± 1.28** | 20.11 ± 1.67 | 22.31 ± 0.98 |
| GRAPHARM-GCN | 70.58 ± 1.59 | 82.71 ± 2.04 | 12.25 ± 1.26 | 12.53 ± 1.01 | 73.25 ± 2.01 | 65.21 ± 1.61 | 16.98 ± 1.21 | 18.64 ± 1.38 |
| GSDM-GCN | 72.21 ± 1.03 | 80.91 ± 0.97 | 11.23 ± 1.11 | 11.78 ± 0.77 | **76.88 ± 0.79** | 67.71 ± 1.59 | 19.58 ± 2.16 | 20.31 ± 1.15 |
| FairAdj-GCN | 66.23 ± 1.12 | 75.67 ± 1.54 | 9.87 ± 0.56 | 10.21 ± 1.81 | 70.93 ± 1.59 | 60.32 ± 1.23 | 14.25 ± 1.51 | 15.48 ± 1.09 |
| FG²AN-GCN | 67.23 ± 1.47 | 74.98 ± 1.32 | 10.84 ± 0.83 | 11.75 ± 1.39 | 71.82 ± 1.79 | 59.48 ± 0.98 | 15.16 ± 1.27 | 16.35 ± 1.90 |
| FairGen-GCN | 69.32 ± 1.87 | 77.31 ± 2.11 | 8.23 ± 1.83 | 9.36 ± 1.52 | 73.43 ± 1.77 | 63.71 ± 1.87 | 12.11 ± 1.19 | 12.81 ± 1.54 |
| FairWire-GCN | 70.81 ± 1.73 | 75.36 ± 1.47 | 9.71 ± 0.88 | 10.33 ± 1.65 | 74.98 ± 1.01 | 61.11 ± 1.09 | 12.98 ± 1.36 | 14.01 ± 2.10 |
| GSDM-GCN | 71.21 ± 0.98 | 75.67 ± 1.08 | 10.02 ± 1.11 | 11.87 ± 1.21 | 74.98 ± 1.01 | 61.11 ± 1.09 | 12.98 ± 1.36 | 14.01 ± 2.10 |
| **FairGEM-GCN** | 72.25 ± 1.33 | 78.41 ± 1.46 | **7.38 ± 1.04** | **8.61 ± 1.28** | 75.25 ± 1.81 | 65.39 ± 1.03 | **9.23 ± 0.59** | **11.46 ± 1.12** |

**Additional results for ablation study.** Figure 3 presents supplementary ablation study results on additional datasets. These results consistently show that each component of FairGEM plays a crucial role in achieving both high generation quality and fairness. Specifically, removing either the fair graph structure regularizer, the fair node feature regularizer, or the disentanglement component leads to noticeable degradation in performance and fairness. These findings confirm the necessity and complementarity of each component within FairGEM for effectively generating high-quality, unbiased synthetic graph.

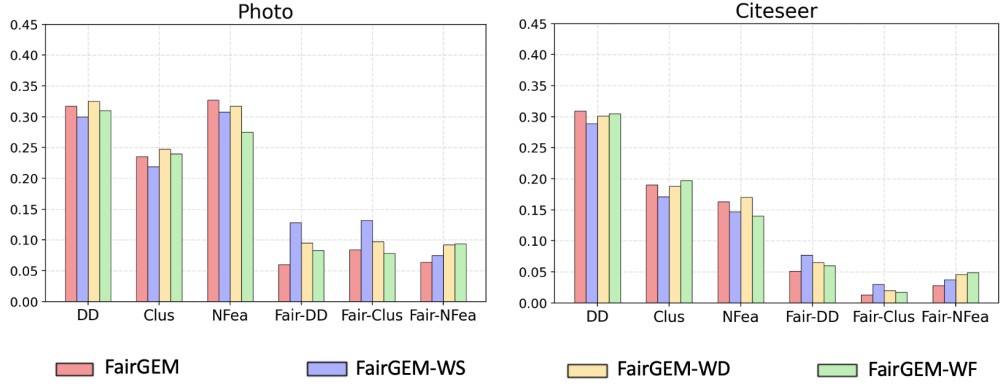

Figure 3: Ablation study results for FairGEM, FairGEM-WS, FairGEM-WD and FairGEM-WF in Photo and citeseer datasets.

