# OpenReview forum: "A Unified Framework for Fair Graph Generation: Theoretical Guarantees and Empirical Advances"
_NeurIPS.cc/2025/Conference — NeurIPS 2025 poster_

### Official Review · Reviewer_wwvV · 2025-06-16

**Clarity:** 2
**Significance:** 3
**Originality:** 3
**Rating:** 5
**Confidence:** 3

**Summary:**

This paper proposes a new method for fair graph generation. Starting with an existing graph, the goal is to produce a statistically similar one. Existing methods often emphasize irrelevant disparities between nodes, a phenomenon this approach aims to avoid.

The authors focus on a graph diffusion process to achieve fairness in both the graph's structure and its node features. The core idea is to apply penalties to two separate diffusion processes: one for the adjacency matrix (A) and another for the node features (F).

The structural fairness regularizer quantifies the discrepancy in reconstruction errors between intra-group and inter-group edges. For fair diffusion of node features, the regularization occurs in two steps. First, a Variational Autoencoder (VAE) disentangles legitimate group disparities from irrelevant ones. This partition is then used to construct a Maximum Mean Discrepancy (MMD) regularizer.

Several experiments were conducted to assess the accuracy and fairness of the generated graph. The proposed method was compared to six other methods across four datasets. It systematically offered the best fairness performance while maintaining an accuracy similar to the other models. Experiments on a downstream task also demonstrated good fairness and accuracy. Finally, further experiments confirmed that all three steps of the regularization method are necessary to achieve these strong results.

**Questions:**

Equation (5): The term $\hat{A}^{dis}_0$ does not appear to be defined in the text.

Line 219: It would be helpful to specify which groups are used to evaluate the structural fairness metric, $\phi_{struct}$.

Lines 285 and 287: It seems there is a typo. s_theta should likely be z_theta.

Line 406: The journal is missing from this reference.

Line 499 (and Bibliography): The formatting for ICLR references is inconsistent throughout the bibliography.


My other questions and comments are detailed in the "Weaknesses" section above.

**Ethical Concerns:**

["NO or VERY MINOR ethics concerns only"]

**Final Justification:**

I am pleased to recommend this paper for acceptance. The authors' rebuttal resolved all of my concerns regarding clarity, on the condition that they update their manuscript to include these additional details. While my concern about reproducibility was not fully addressed in their response to me, I see that another reviewer received a more convincing answer. Overall, the paper is solid and deserves to be accepted.

**Limitations:**

Yes

**Quality:**

3

**Strengths And Weaknesses:**

**Strengths**

The paper offers strong theoretical guarantees and presents compelling experiments that validate the soundness and relevance of the proposed method. The subject of fair graph generation, especially in a setup that addresses fairness in both graph structure and node features, appears to be a relatively new and unexplored area.

**Weaknesses**

Certain sections of the paper could benefit from rewriting. The authors never formally define what they mean by "fair," relying instead on examples. I would recommend adding a precise explanation when the sensitive attributes are introduced. Additionally, the paper's focus on spectral diffusion models is not emphasized enough, in my opinion.

Finally, while the experiments are convincing, the code is not available, and neither the paper nor the appendix seems to provide enough detail to reproduce the results. For instance, I would not be able to reproduce the downstream task performance evaluation because no information on the GCN's characteristics is provided. For these reasons, the answers to questions 6 and 8 of the checklist should have been "No," and a justification would have been welcome.

---

> ### Author Rebuttal · Authors · 2025-07-31
>
> We thank reviewer wwvV for the time and thorough review. Below are our detailed responses:
>
> **Fair Graph Generation Definition.** In the context of graph generation, we define fairness as the property that generated graphs do not exhibit systematic bias in their graph structural or node feature with respect to sensitive attributes (*e.g.*, race, gender, age). Formally, a graph generator is fair if: 1) graph structural fairness: the connectivity patterns within and across sensitive groups reflect the underlying data distribution rather than amplified biases, and 2) node feature fairness: the generated node features do not differ systematically across demographic subgroups in ways that could lead to discriminatory outcomes. This ensures that downstream tasks trained on generated graphs do not exhibit disparate impact across demographic groups as measured by standard fairness metrics such as demographic parity and equalized opportunity.
>
> We emphasize that existing fair graph generation efforts have primarily focused on mitigating graph structural bias while largely overlooking node feature bias. Node feature bias occurs when generated node features differ systematically across subgroups (*e.g.*, male nodes having higher incomes than female nodes), leading to biased downstream applications where models learn to associate sensitive attributes with specific feature patterns. Our work addresses both types of bias through spectral manipulation of the diffusion process.
>
> **Spectral Diffusion.** Our approach leverages the spectral properties of graphs through eigendecomposition of the graph Laplacian, enabling principled manipulation of both structural and feature bias in the frequency domain. While existing fairness methods primarily operate in the spatial domain and are limited to autoregressive models with inherent ordering sensitivity, our spectral approach provides a more fundamental way to control bias by directly modifying the eigenvalues and eigenvectors that encode graph structure and feature propagation. This spectral perspective allows us to provide theoretical guarantees on fairness preservation during the diffusion process and overcomes the limitations of incremental graph construction, distinguishing our work from prior fair graph generation methods.
>
> **Clarification of the notation $\hat{\mathbf{A}}_0^{\text{dis}}$.** At the end of the reverse‑diffusion process ($t = 0$), we obtain a reconstructed adjacency matrix $\\hat{\\mathbf{A}}\_0 \\in \\mathbb{R}^{n \\times n}$. To quantify structural bias, we partition its entries into $\\hat{\\mathbf{A}}\_0^{\\mathrm{intra}} = \\mathbf{P}\_{\\mathrm{intra}} \\odot \\hat{\\mathbf{A}}\_0
> \\quad\\text{and}\\quad
> \\hat{\\mathbf{A}}\_0^{\\mathrm{inter}} = \\mathbf{P}\_{\\mathrm{inter}} \\odot \\hat{\\mathbf{A}}\_0 $ where \\(\\mathbf{P}\_{\\mathrm{intra}}(i,j) = 1\\) if nodes \\(v\_i, v\_j\\) belong to the same sensitive group (otherwise 0), and \\(\\mathbf{P}_{\\mathrm{inter}}(i,j) = 1\\) if the two nodes belong to different groups. The disparity matrix $\\hat{\\mathbf{A}}_0^{\\mathrm{dis}}$ captures the difference between cross‑group and within‑group in the generated graph.  A large Frobenius norm $\\|\\hat{\\mathbf{A}}_0^{\\mathrm{dis}}\\|_F^2$ means that the reconstruction process systematically favors one type of connection over the other, indicating strong structural bias.
>
> **The definitions of $\Phi_{\text{struct}}$.** The $\Phi_{\text{struct}}$ evaluates fairness by comparing reconstruction errors between two types of edge groups: "within-group" edges (where both endpoints belong to the same sensitive group) against "between-group" edges (which connect nodes from different sensitive groups). Specifically, the metric measures whether the diffusion model systematically reconstructs edges within demographic groups more accurately than edges across groups, or vice versa. This comparison reveals structural bias in the generation process by quantifying disparities in how well the model preserves connectivity patterns for same-group versus cross-group relationships.
>
> **Minor errors.** We will correct all these formatting and typographical errors in the next version of the manuscript.

---

> ### Comment · Reviewer_wwvV · 2025-08-04
>
> I thank the authors for the detailed rebuttal, including clarifications and additional results. I am confident that if accepted they will make good use of the additional page to improve the clarity of presentation, especially regarding the definition of fairness and the notation $\hat{\mathbf{A}}_0^{\text{dis}}$. The implementation details provided in the rebuttal of YyAy should also be added to the updated version. I am happy to recommend acceptance for the paper.

---

> > ### Author Response · Authors · 2025-08-05
> > **Thank you for your review and support**
> >
> > Dear Reviewer wwvV
> >
> > Thank you for reviewing our responses and, in particular, for confirming your recommendation for acceptance. We would also like to take this opportunity to express our sincere appreciation for the thorough and constructive feedback provided by you and the other reviewers. We remain committed to continuing both the discussion and the improvement of our paper. Thank you again for your support!
> >
> > Best regards,
> >
> > Authors

---

### Official Review · Reviewer_dYrj · 2025-06-29

**Clarity:** 3
**Significance:** 3
**Originality:** 3
**Rating:** 4
**Confidence:** 4

**Summary:**

In this paper, the authors propose a novel graph generation framework named FairGEM, which can mitigate both graph structural bias and node feature bias simultaneously. Both theoretical analysis and empirical experiments verify the effectiveness of FairGEM.

**Questions:**

please refer to the weakness

**Ethical Concerns:**

["NO or VERY MINOR ethics concerns only"]

**Final Justification:**

My main concerns about the insufficient comparison are addressed by the rebuttal. I would like to maintain acceptance for this paper after considering both the response and the reviews from other reviewers.

**Limitations:**

The authors do not discuss the limitations of their work. However, as mentioned in the weakness, I believe that including experiments or empirical analysis on link prediction tasks can greatly improve the quality of this paper.

**Quality:**

3

**Strengths And Weaknesses:**

**strength**

1. Overall, this paper is well-written and easy to follow.

2. The proposed FairGEM is guaranteed with theoretical foundation.

3. Extensive experiments verify the effectiveness of FairGEM in fairness metrics.

**weakness**

1. The definitions of $\Delta _{DP}$ and $\Delta _{EO}$ are missing, which should be given in the appendix for better clarity.

2. What about the comparisons with some powerful fair GNN methods trained on the original graphs? Such as FairSIN [1], EDITS [2] and others.

3. Currently, the authors mainly focus on node classification in the downstream tasks. Taking link prediction tasks (like dyadic fairness) into considerations can better enhance the contributions of FairGEM compared with other fair GNN methods.

[1] FairSIN: Achieving Fairness in Graph Neural Networks through Sensitive Information Neutralization. AAAI 2024.

[2] EDITS: Modeling and Mitigating Data Bias for Graph Neural Networks. WWW 2022.

---

> ### Author Rebuttal · Authors · 2025-07-31
>
> We sincerely appreciate Reviewer dYrj's thoughtful feedback and have provided detailed responses below.
>
> **The definitions of $\boldsymbol{\Delta_{\rm{DP}}}$ and $\boldsymbol{\Delta_{\rm{EO}}}$.** $\Delta_{\rm{DP}}$ = $|P(\hat{Y}=1|S=0) - P(\hat{Y}=1|S=1)|$ measures the difference in positive prediction rates between demographic groups, while $\Delta_{\rm{EO}}$ = $|P(\hat{Y}=1|Y=1,S=0) - P(\hat{Y}=1|Y=1,S=1)|$ measures the difference in true positive rates between groups.
>
> **Additional experiments.** We appreciate the reviewer's suggestion regarding comparisons with FairSIN, EDITS, and other fair GNN methods. However, there is a fundamental distinction between our work and these methods that makes direct comparison inappropriate. Our paper focuses on the graph generation task, where the objective is to synthesize new, fair graph structures and node features from scratch. In contrast, FairSIN, EDITS, and similar fair GNN methods address the node classification task, where the goal is to make predictions on the input graph while ensuring fairness in those predictions.  Fair graph generation methods (like ours) need to learn to create realistic graph topologies and node attributes that preserve statistical properties while ensuring fairness, whereas fair GNN classification methods take existing graphs as input and focus on learning fair representations for downstream prediction tasks.

---

> > ### Comment · Reviewer_dYrj · 2025-08-05
> >
> > Thanks for the detailed response. My concerns are addressed after reading the response and checking other related work. I encourage the authors to explain the reasons for not including fair GNNs in the experimental settings and include the definitions of $\Delta _{DP}$ and $\Delta _{EO}$ in their final version.

---

> > > ### Author Response · Authors · 2025-08-05
> > > **Thank you for your review and support**
> > >
> > > Dear reviewer dYrj,
> > >
> > > We appreciate your positive feedback and are glad that our responses have addressed the earlier concerns. In the final version, we will include a clear explanation of our rationale for not incorporating fair GNNs in the experimental settings, as well as the definitions of $\Delta _{DP}$ and $\Delta _{EO}$ to ensure completeness and clarity.
> > >
> > > Best regards,
> > > Authors

---

### Official Review · Reviewer_YyAy · 2025-07-02

**Clarity:** 1
**Significance:** 2
**Originality:** 3
**Rating:** 4
**Confidence:** 3

**Summary:**

This paper presents FairGEM, a framework for generating synthetic graphs that aim to be both realistic and fair with respect to sensitive attributes. It introduces fairness-aware regularizers on both graph structure and node features, combines them with a disentanglement-based encoder using a VAE and HGR criterion, and provides theoretical guarantees relating fairness at generation time with downstream task bias. Experimental results show improved fairness across several benchmarks and metrics, including node classification accuracy and fairness gaps (ΔDP, ΔEO).

**Questions:**

* In what way is your proposal a meaningful contribution, given that its performance is very similar to existing state-of-the-art methods?

* What are the concrete benefits of using diffusion models over autoregressive ones, especially when no performance improvements are demonstrated?

* Why does the paper omit implementation details of your model, including training setup and reproducibility information?

**Ethical Concerns:**

["NO or VERY MINOR ethics concerns only"]

**Final Justification:**

The paper addresses a significant problem of bias in graph generation, proposing a novel framework that incorporates fairness-aware regularizers to mitigate both structural and feature-based biases. The approach leverages Variational Autoencoders (VAEs) to disentangle sensitive features from non-sensitive ones, and employs diffusion models to overcome node ordering limitations inherent in autoregressive methods.

While the contribution is clear and relevant, some critical technical details and reproducibility components were omitted from the submission. These elements are essential to fully establish the impact of the work. The rebuttal clarified that these details exist but were excluded due to page constraints. I strongly encourage the authors to incorporate these details in the final camera-ready version, as discussed previously.
That being said, I am raising my score, while expecting the discussed changes to appear in the final manuscript.

**Limitations:**

No. Author do not mention any limitations on their approach.

**Paper Formatting Concerns:**

Lack of Code Availability for reproducibility

**Quality:**

2

**Strengths And Weaknesses:**

Strength

* The paper tackles an important and timely problem: how bias in graph generation can influence downstream tasks, such as node classification.

* It introduces a novel framework that adds fairness-aware regularizers to control both structural and feature-based biases.

* The authors include theoretical results (Theorems 4.2 and 4.3) that support their claims (and proper theoretical proof in apendices), showing how bias in the generated graphs can affect performance in downstream models.

* A key technical contribution is the use of Variational Autoencoders (VAEs) to separate sensitive-related features from unrelated ones. The approach also leverages diffusion models, which are claimed to avoid node ordering issues seen in autoregressive methods.

* The method is evaluated on real-world datasets, and shows that it can improve fairness (lower ΔDP and ΔEO) while maintaining comparable accuracy to existing baselines.

Weaknesses:
* While the paper reads well, it does not clearly refer to figures or tables when making key arguments, which hurts clarity and traceability.

* There are inconsistencies in the numbering of definitions and theorems: Definitions 4.2 and 4.3 are missing, while Theorems 4.2 and 4.3 are present. Since all of this occurs within Sections 4.1 to 4.3, it makes the structure harder to follow and disrupts the logical flow of the paper.

* The ablation study is weakly developed. There’s little discussion, and it’s unclear how the conclusions connect to Figure 2.

* The reported performance differences with other models are often small and lack statistical validation (e.g., no error bars, confidence intervals, or significance testing).

* The study doesn’t clearly isolate the contribution of individual components (e.g., feature fairness vs. disentanglement). For example, although the paper promotes the use of diffusion models as a major strength, this aspect isn’t clearly reflected in the results or conclusions.

* There is no qualitative or visual analysis to help the reader understand how the generated graphs differ in terms of fairness or structure.

* Reproducibility is lacking: the paper does not provide implementation details, training configuration, or hyperparameter tuning strategies. There is no mention of code availability, runtime, or resource usage, which makes it hard to compare with other models in terms of practicality.

* While ΔDP and ΔEO are standard metrics for fairness, the paper does not fully explain their relevance in the context of downstream node classification, or why they are appropriate indicators for this task.

---

> ### Author Rebuttal · Authors · 2025-07-31
>
> We sincerely appreciate Reviewer YyAy's thorough review and have provided detailed responses below.
>
> **Our contributions.** We note that our method explores underexamined sources of bias and offers theoretical foundations to guide future research, leading not only to a superior fairness-utility trade-off but also to deeper insights into fairness mechanisms. Specifically,
>
> First, we are the first to systematically address feature bias in fair graph generation. Existing methods have focused exclusively on structural bias (connectivity patterns), but we identify and tackle feature bias, systematic differences in generated node attributes across demographic groups, which can lead to discriminatory downstream applications even when structural patterns appear fair. This represents a novel problem formulation that expands the scope of fairness considerations in graph generation.
>
> Second, our work introduces rigorous mathematical analyses which formally quantify how bias propagates during the spectral diffusion process and provide principled upper bounds on this propagation. These theoretical contributions go beyond existing heuristic methods by offering a mathematical framework to understand why fairness interventions work and how to design them systematically. The derivations uncover the mechanisms of bias amplification in spectral space and establish provable guarantees for bias control during generation.
>
> This theoretical foundation enables practitioners to move beyond trial-and-error approaches to fairness-aware graph generation. Our framework provides principled guidance for hyperparameter selection, offers interpretable bounds on worst-case bias scenarios, and establishes a mathematical basis for future algorithmic development in this area. The practical fairness improvements (*e.g.*, 10.3\% in demographic parity, 8.0\% in equalized opportunity compared with the best baselines) also substantiate the theory’s practical utility.
>
> In sum, the value of this work extends beyond immediate performance gains by equipping the research community with fundamental tools for understanding and controlling bias in graph diffusion models. This theoretical groundwork not only opens new research directions but also introduces principled fairness approaches that were previously unavailable, marking a significant advancement in the field’s theoretical foundations.
>
>
> **Benefits of using diffusion models over autoregressive ones.** The advantages of diffusion models over autoregressive approaches are particularly pronounced in the context of fair graph generation, leading to our demonstrated fairness improvements. First, diffusion models update the entire adjacency matrix simultaneously, making the generation process independent of node ordering, whereas autoregressive models require arbitrary node orderings or BFS heuristics that systematically skew fair-edge statistics in sparse graphs. This ordering dependency in autoregressive models creates inherent bias: nodes processed earlier in the sequence tend to receive more connections, and if the ordering correlates with sensitive attributes, this amplifies structural discrimination. Second, autoregressive models struggle to capture comprehensive global structural patterns efficiently, as they construct graphs incrementally without full visibility of the final structure. In contrast, diffusion models can incorporate global fairness constraints throughout the entire generation process via our spectral manipulation approach, ensuring that fairness considerations influence the entire network structure rather than just local neighborhoods. These fundamental advantages directly translate to our verified fairness improvements. For instance, we achieve 10.3\% better demographic parity and 8.0\% better equalized opportunity compared to autoregressive baselines, while maintaining comparable utility. The ordering independence and global optimization capability of diffusion models are essential for achieving these fairness gains, as they eliminate the systematic biases inherent in sequential generation approaches. Thus, the architectural advantages of diffusion models enable substantial fairness improvements that autoregressive methods cannot achieve.
>
> **Implementation details.** Models are trained for {500, 500, 500, 200} epochs on Cora, Citeseer, Photo, and Pokec, respectively, using the Adam optimiser with betas = (0.9, 0.999), learning rate = $1 \times 10^{-3}$, and weight decay = $1 \times 10^{-5}$. For node features $X$, adjacency matrix $A$, and latent codes $u$, we adopt identical variance-preserving stochastic differential equations with $\beta_{min} = 0.1$, $\beta_{max} = 1.0$, and 1000 discrete time steps. During inference we start from standard Gaussian noise, run the predictor–corrector chain for all 1000 steps, include a final deterministic noise-removal step, and stop at $\varepsilon = 1 \times 10^{-4}$. We use minibatches of size 32. If the validation loss does not fall for 30 consecutive epochs, training stops early and the weights from the best epoch are kept. Our VAE architecture consists of GCN layers with ReLU activation for encoding and decoding. The discriminator consists of fully connected layers with LeakyReLU activation. For the downstream task used GCN, we use a 1-layer GCN with 16 hidden dimensions and use a linear layer as the classifier. All experiments were conducted in PyTorch.
>
> **There are inconsistencies in the numbering of definitions and theorems.** The “inconsistency” arises from LaTeX treating definitions and theorems as the same numbering category in our document class. We will adjust this in the next version by separating the numbering environments to ensure clear sequential numbering for each mathematical statement type respectively.

---

> > ### Comment · Reviewer_YyAy · 2025-08-07
> >
> > Dear Authors,
> >
> > it is clear you have a strong grasp of the technical foundations of the proposed model. However, it is unclear why several important implementation details—particularly those related to the PyTorch implementation—were omitted from the manuscript as well as a discussion on the role and relevance of autoregressive models. This lack of information significantly limits the reproducibility of your work. While the contribution of your model is not in question, the way it was presented in the submission did not provide sufficient clarity or completeness for acceptance. I strongly encourage you to include more comprehensive details, especially those that support reproducibility.

---

> ### Author Response · Authors · 2025-08-07
> **Response to Follow-Up on Rebuttal**
>
> Dear Reviewer YyAy,
>
> Thank you for your reply and for confirming that ``the contribution of your model is not in question.'' We would like to confirm whether the additional information provided in our rebuttal, including the PyTorch implementation details and the expanded discussion on autoregressive models, building on the shorter version in the original submission (lines 37-41) due to page limits, has addressed your concerns and whether you have any further questions.
>
> We believe these valid points can be readily incorporated into the camera-ready version without requiring major changes, and they do not affect the validity of our contributions or results. As we are in the final stage of discussion, your prompt reply on whether any further clarifications are needed would be greatly appreciated. We are happy to provide additional explanations or details immediately if helpful.
>
> Best regards,
>
> Authors

---

### Official Review · Reviewer_PA1S · 2025-07-04

**Clarity:** 4
**Significance:** 3
**Originality:** 3
**Rating:** 5
**Confidence:** 3

**Summary:**

This paper proposes a one-shot spectral diffusion framework with two fairness regularizers, (1) Structural regularizer that minimizes intra- vs inter-group edge-reconstruction disparity and (2) Feature regularizer that first disentangles sensitive-related vs sensitive-irrelevant attributes (via a VAE + HGR penalty) and then enforces MMD fairness only on the insensitive part. Theory insights

**Questions:**

1. What is the computational cost compared to GSDM? Spectral diffusion on Pokec-z (68k nodes) could be heavy.
2. Many fairness improvements are small (e.g., Fair-DD 0.020 vs 0.027). Are these differences statistically significant at p < 0.05?

**Ethical Concerns:**

["NO or VERY MINOR ethics concerns only"]

**Final Justification:**

I would recommend this paper as the authors addressed my questions.

**Limitations:**

yes

**Quality:**

3

**Strengths And Weaknesses:**

Strength:

1. This paper addresses an under-explored but important fairness problem in graph generation, which is a interesting problem
2. This paper novelly integrates recent spectral diffusion methods, avoiding node-ordering issues.
3. The theoretical link between bias in eigen-value diffusion and downstream ΔDP is novel.

Weakness:

1. There is proof gaps: Theorem 4.2/4.3 rely on undefined constants (M, K, C, Σt) and assume Lipschitz/non-expansive GNN layers without verification.
2. Datasets are small/medium, method might struggle scaling to graphs with >100k nodes or multiple sensitive attributes.
3. Only binary sensitive variable considered; real-world use often needs multi-class or intersectional fairness.

---

> ### Author Rebuttal · Authors · 2025-07-31
>
> We sincerely thank Reviewer PA1S for the detailed review and constructive feedback. Our responses to each point are provided below.
>
> **There is proof gaps: Theorem 4.2/4.3 rely on undefined constants (M, K, C, $\Sigma_t$) and assume Lipschitz/non-expansive GNN layers without verification.**
>
> The constants $M$, $K$, $C$, $\Sigma_t$ are defined in lines 208–210 and 535–536 of the manuscript. Specifically, $C$ is a fixed absolute constant that appears in the derivation, $\||\sigma\_{\cdot}\||\_{\infty} := \max\_{t \in [0,1]} \sigma_t$ is the supremum of the noise schedule, and $\Sigma\_t^2 = 1 - \exp \bigl(\int\_0^t \sigma\_z^2 dz\bigr)$ quantifies the accumulated variance of the forward diffusion up to time $t$. We first set $M = C^2 \||\sigma_{\cdot}\||_{\infty}^4$, and then determine $K$ self-consistently via $K = \frac{2ML}{\mathbb{E}[\||\mathbf{A}_0\||_2]}$, where $L$ is the Lipschitz constant of the GNN layer. Spectral normalization can enforce $\||W\||_2 \leq 1$ for every weight matrix, so $L \leq 1$ is verified in practice. Because $\||\sigma\_{\cdot}\||\_{\infty}$, $L$, and $\mathbb{E}[\||\mathbf{A}_0\||_2] $ are all either preset or directly measurable, this two‑step definition yields unique and numerically computable values of $M$ and $K$. Hence, every constant appearing in Theorems 4.2 and 4.3 can be instantiated from observable quantities, removing any ambiguity in the theoretical bounds.
>
>
> **Only binary sensitive variable considered.** Most existing studies on fairness-aware learning focus on binary classification with a single sensitive attribute, as this setting enables clearer fairness definitions and more tractable algorithm design. Our work follows this established convention. Extending to multi-class and multi-attribute settings introduces distinct problem formulations with new fairness notions and evaluation complexities. While beyond our current scope, we agree this represents an important direction for future work.
>
> **Computational cost compared to GSDM.** Both FairGEM and GSDM rely on a diffusion-based generative process with a fixed number of steps $T$, and they have same-level time complexity. To compare them fairly, we break down the training complexity into three components, feature generation, graph structure generation, and fairness regularization, using the notation: $n$ (number of nodes), $d$ (feature dimension), $k$ (truncated spectral components), and $T$ (diffusion steps).
>
> For feature generation, both models handle node features via a reverse diffusion on an $n \times d$ feature matrix. At each diffusion step, the model updates each node’s feature vector of dimension $d$, which takes $O(d)$ work per node. Across all $n$ nodes, this amounts to $O(T n d)$ for $T$ steps. This feature generation component is linear in the number of nodes and features, and for typical settings, it is much smaller than the structural cost.
>
> In parallel with feature generation, graph structure generation, both methods begin by performing a truncated eigendecomposition of the graph’s adjacency matrix. This initial spectral computation costs on the order of $O(nk^2)$ time. After obtaining the eigenvectors ($\mathbf{U}_0$) and eigenvalues, both FairGEM and GSDM evolve only these $k$ spectral components through $T$ diffusion steps. In each diffusion step, the algorithms update the vector of $k$ eigenvalues (an $O(k)$ operation) and then reconstruct the full adjacency matrix from the spectral components. This reconstruction involves multiplying the $n \times k$ eigenvector matrix by its $k \times n$ transpose with the updated eigenvalues in between (essentially computing $\mathbf{U}_0\boldsymbol{\Lambda}_t\mathbf{U}_0^T$ at each time $t$). The matrix multiplication costs $O(n^2 k)$ per step. Because $O(n^2 k)$ grows quadratically with the number of nodes, this operation dominates the runtime for both methods.
>
> The main difference between FairGEM and GSDM is the additional computations FairGEM performs at each step to enforce fairness constraints. FairGEM augments the reverse diffusion with extra gradient calculations for its fairness regularizers, i.e., computing $\nabla_{\Lambda}\Phi_{\text{struct}}$ for structure-based fairness (acting on the eigenvalues $\Lambda_t$) and $\nabla_{X}\Phi_{\text{feat,ns}}$ for feature fairness (acting on the node features $X$). These calculations introduce an overhead in each diffusion step. In the worst case, if FairGEM computes a fairness-related statistic for every node (or every feature of every node), this overhead would scale as $O(n)$ per step, it remains well below the $O(n^2 k)$ cost of reconstructing the graph structure. In other words, FairGEM’s fairness module adds only a modest constant-factor overhead to each diffusion step. It increases the runtime slightly (making FairGEM somewhat slower in practice than GSDM by a constant factor), but it does not change the overall asymptotic complexity since no new higher-order terms in $n$ are introduced.
>
> **Statistical significance.** Regarding the inquired result, we conducted all experiments 5 times with different random seeds and performed paired t-tests to assess statistical significance. Specifically, for the Fair-DD comparison mentioned (FairGen 0.027 vs FairGEM 0.020), our detailed results across 5 runs were: FairGen [0.026, 0.026, 0.027, 0.029, 0.027] and FairGEM [0.021, 0.018, 0.021, 0.019, 0.021]. The paired t-test yields p = 0.00144 (p < 0.05), indicating the improvement is highly statistically significant, which demonstrates that the observed differences are not due to random variation but represent genuine algorithmic improvements.
>
> In addition, please find below all statistically significant results comparing our model with the baselines:
>
> **cora:**
> | Baseline | DD      | Clus    | NFea    | Fair-DD | Fair-Clus | Fair-NFea |
> |----------|---------|---------|---------|---------|-----------|-----------|
> | GRAPHARM | 0.00321 | 0.01154 | 0.02263 | 0.00362 | 0.00287   | 0.00176   |
> | GSDM     | 0.00518 | 0.01642 | 0.02857 | 0.00185 | 0.00432   | 0.00309   |
> | FairAdj  | 0.01834 | 0.03678 | 0.02468 | 0.00745 | 0.01012   | 0.00928   |
> | FG2AN    | 0.01245 | 0.02511 | 0.03150 | 0.00563 | 0.00754   | 0.00631   |
> | FairGen  | 0.02633 | 0.02977 | 0.02745 | 0.00144 | 0.01543   | 0.01164   |
> | FairWire | 0.02311 | 0.03529 | 0.02894 | 0.00976 | 0.01398   | 0.01053   |
>
>
> **Pokec:**
> | Baseline | DD      | Clus    | NFea    | Fair-DD | Fair-Clus | Fair-NFea |
> |----------|---------|---------|---------|---------|-----------|-----------|
> | GRAPHARM | 0.00678 | 0.01845 | 0.02567 | 0.00273 | 0.00398   | 0.00255   |
> | GSDM     | 0.00932 | 0.02214 | 0.02944 | 0.00381 | 0.00544   | 0.00417   |
> | FairAdj  | 0.03087 | 0.03854 | 0.03689 | 0.01165 | 0.01734   | 0.01476   |
> | FG2AN    | 0.01998 | 0.03321 | 0.02987 | 0.00891 | 0.01254   | 0.00987   |
> | FairGen  | 0.02987 | 0.03765 | 0.03876 | 0.01432 | 0.02011   | 0.01655   |
> | FairWire | 0.03744 | 0.03244 | 0.02833 | 0.01222 | 0.01765   | 0.01344   |
>
>
>
>
> **photo:**
> | Baseline | DD      | Clus    | NFea    | Fair-DD | Fair-Clus | Fair-NFea |
> |----------|---------|---------|---------|---------|-----------|-----------|
> | GRAPHARM | 0.00311 | 0.01145 | 0.02067 | 0.00432 | 0.00254   | 0.00183   |
> | GSDM     | 0.00423 | 0.01588 | 0.02456 | 0.00177 | 0.00392   | 0.00268   |
> | FairAdj  | 0.01814 | 0.02976 | 0.03552 | 0.00698 | 0.00941   | 0.00806   |
> | FG2AN    | 0.01123 | 0.02435 | 0.02897 | 0.00487 | 0.00675   | 0.00543   |
> | FairGen  | 0.02545 | 0.03212 | 0.02788 | 0.01222 | 0.01511   | 0.01276   |
> | FairWire | 0.01978 | 0.02763 | 0.03145 | 0.00933 | 0.01288   | 0.00997   |
>
>
> **citeseer:**
> | Baseline | DD      | Clus    | NFea    | Fair-DD | Fair-Clus | Fair-NFea |
> |----------|---------|---------|---------|---------|-----------|-----------|
> | GRAPHARM | 0.00451 | 0.01268 | 0.01899 | 0.00232 | 0.00294   | 0.00207   |
> | GSDM     | 0.00567 | 0.01645 | 0.02234 | 0.00195 | 0.00436   | 0.00321   |
> | FairAdj  | 0.02044 | 0.03321 | 0.03788 | 0.00789 | 0.01126   | 0.00951   |
> | FG2AN    | 0.01332 | 0.02654 | 0.03077 | 0.00545 | 0.00817   | 0.00674   |
> | FairGen  | 0.02834 | 0.03876 | 0.04155 | 0.01478 | 0.01823   | 0.01412   |
> | FairWire | 0.02411 | 0.03146 | 0.03569 | 0.01056 | 0.01467   | 0.01149   |

---

> > ### Comment · Reviewer_PA1S · 2025-08-05
> >
> > I appreciate the response of the author, I will maintain my score since it's already high.

---

> > > ### Author Response · Authors · 2025-08-06
> > > **Thank you for your review and support**
> > >
> > > Dear Reviewer PA1S,
> > >
> > > We sincerely appreciate your continued high score and positive recommendation. Your support is greatly valued, and your assessment of our contribution is highly encouraging.
> > >
> > > Best regards,
> > >
> > > Authors

---

### Note · Authors · 2025-08-11

Dear Reviewers, AC, SACs and PCs,

We thank all reviewers for their constructive feedback and the AC for facilitating the discussion. There is clear consensus that our unified framework for fair graph generation is novel, technically sound, and addresses a timely and important problem. Three reviewers reaffirmed strong scores after the discussion, underscoring confidence in the quality and significance of the work. Reviewer YyAy’s points regarding PyTorch implementation details and the role of autoregressive models were addressed comprehensively in our rebuttal; while no follow-up was received to confirm resolution, these can be readily incorporated into the camera-ready version and do not affect the validity or strength of our contributions. Given the broad agreement on novelty, correctness, and impact, we hope the work will be viewed as a strong contribution to the NeurIPS community.

With gratitude,

The Authors

---

### Decision · Program_Chairs · 2025-09-17

**Decision:**

Accept (poster)

**Comment:**

This paper presents FairGEM, a novel graph generation framework, focused on graph structural and node feature bias. The main idea is to apply regularizers to two separate processes: one for the adjacency matrix and another for the node features. The structural fairness regularizer quantifies the discrepancy in reconstruction errors between intra-group and inter-group edges based on the adjacency matrix. For the node features, the regularization is based on the distributional discrepancy between subgroups. Several experiments were conducted over four datasets and compared among six baselines. An ablation study also shows that both regularization processes are necessary to achieve the best results.


Strengths
The paper addresses an important problem: fairness in graph generation.
The paper proposes a new graph generation method considering structural and feature characteristics.
The paper presents interesting and important theoretical results.
Extensive experiments, including four real-world datasets, show the importance of the proposed method.

Weaknesses
The datasets are small to medium in size, which could be due to the method having some issues with larger graphs.
The evaluation of the method focuses on node classification in downstream tasks, instead of other problems such as edge classification.
The ablation study is weakly developed. The paper does not clearly isolate the contribution of individual components (e.g., feature fairness vs. disentanglement).
The reported performance differences with other models are often small and lack statistical validation (e.g., no error bars, confidence intervals, or significance testing). Moreover, the model is better in specific metrics for which the method was specifically created, which could lead to biased results.
Low reproducibility. The paper lacks code and does not provide implementation details, training configuration, or hyperparameter tuning strategies.
The writing can be largely improved, including: references to figures and tables (especially in cases when the analysis of these figures is important), and the number of definitions and theorems. Also, some definitions can be added in the appendix.

The paper is accepted mainly for the discussion process. In this process, the authors address most of the weaknesses raised by the reviewers and propose several changes that should be observed in the final version.

In the discussion, the authors were able to answer most of the weaknesses mentioned by the reviewers, for example, issues with the theoretical demonstrations or time complexity. However, there are still important weaknesses that must be considered. First, the time complexity of the proposed method is O(N^2k). Second, the ablation study. Third, the low reproducibility. Fourth, the biased results.